# Exploring the Precise Dynamics of Single-Layer GAN Models: Leveraging Multi-Feature Discriminators for High-Dimensional Subspace Learning

**Andrew Bond**
KUIS AI Center
Koç University
abond19@ku.edu.tr

**Zafer Doğan** [*]
MLIP Research Group, KUIS AI Center
Electrical and Electronics Engineering
Koç University
zdogan@ku.edu.tr

## Abstract

Subspace learning is a critical endeavor in contemporary machine learning, particularly given the vast dimensions of modern datasets. In this study, we delve into the training dynamics of a single-layer GAN model from the perspective of subspace learning, framing these GANs as a novel approach to this fundamental task. Through a rigorous scaling limit analysis, we offer insights into the behavior of this model. Extending beyond prior research that primarily focused on sequential feature learning, we investigate the non-sequential scenario, emphasizing the pivotal role of inter-feature interactions in expediting training and enhancing performance, particularly with an uninformed initialization strategy. Our investigation encompasses both synthetic and real-world datasets, such as MNIST and Olivetti Faces, demonstrating the robustness and applicability of our findings to practical scenarios. By bridging our analysis to the realm of subspace learning, we systematically compare the efficacy of GAN-based methods against conventional approaches, both theoretically and empirically. Notably, our results unveil that while all methodologies successfully capture the underlying subspace, GANs exhibit a remarkable capability to acquire a more informative basis, owing to their intrinsic ability to generate new data samples. This elucidates the unique advantage of GAN-based approaches in subspace learning tasks.

## 1 Introduction

Subspace learning is a widely explored task, especially with the growth of dimensionality in modern datasets. It is important to identify meaningful subspaces within the data, such as those determined by principal component analysis (PCA). However, due to the high dimensionality of the data, it is common to employ online methods such as Oja's method [1] and GROUSE [2]. Meanwhile, Generative Adversarial Networks (GANs) [3], primarily used as generative models, have also demonstrated the ability to learn meaningful representations of data [4, 5]. Inspired by this, we explore how single-layer GAN models can be viewed as a form of subspace learning.

We seek to improve the understanding of GAN training by relaxing some common assumptions made in previous analysis of GANs [6]. Specifically, we focus on the training dynamics of the gradient-based learning algorithms, which can be converted into a continuous-time stochastic process characterized by an ordinary differential equation (ODE). Furthermore, the dynamics of the model weights form a stochastic process modeled by a stochastic differential equation (SDE). Understanding these two equations provides the relevant information to understand convergence behaviour of the

---

[*]Corresponding Author

38th Conference on Neural Information Processing Systems (NeurIPS 2024).

training. We extend previous work to discriminators with same dimensionality as the generator. Finally, we discuss the new training outcomes that can arise in the higher-dimensionality case.

Our work explores what happens when the training switches from sequential learning in the single-feature discriminator case [6], to the non-sequential (multi-feature) learning of our discriminator. We show that the non-sequential learning of features not only allows for faster learning and convergence, but also a higher maximum similarity with the true subspace compared to the sequential case, when everything else is kept the same. This shows that contrary to the general approach of making discriminators much weaker than generators, it is still possible to use a powerful discriminator. In fact, doing so can lead to much faster training with better performance, through careful choice of learning rates.

We further show that our new framework can be used to analyze the cases where we assume different dimensionalities between the true subspace, fake subspace, and discriminator. Through the use of a simple uplifting trick on the relevant Grassmannians, we are able to extend our analysis to arbitrary dimensionalities. To understand how GANs compare with existing subspace learning algorithms, we provide both theoretical and empirical comparisons with existing such algorithms. We see that the features learned by a GAN model are more meaningful and represent the data better as compared to Oja's method, due to the requirement of being able to generate new data from the underlying data distribution.

Finally, we test our approach using two prominent real-world datasets: MNIST and Olivetti Faces, comparing our method against a sequential discriminator, and show that all of the key insights gained through the theoretical analysis are visible in training on this dataset as well. This shows that our analysis has very practical applications, and can lead to interesting research directions exploring these ideas in more powerful GAN architectures. This testing is additionally done on the case where we assume different dimensionalities for each component of the model, showing that the results are as expected. We release all our code at `https://github.com/KU-MLIP/SolvableMultiFeatureGAN`.

Overall, our contributions are as follows:

1. We investigate the distinctions between multi-feature and single-feature discriminators and fully characterize the learning dynamics through a rigorous scaling limit analysis.

2. We introduce a novel method for analyzing cases where the true feature dimensionality is unknown, enabling broader future analyses under uncertain conditions.

3. We position the multi-feature GAN models as a new type of subspace learning algorithm, and compare against existing algorithms, both theoretically and empirically.

4. We further validate our findings on image datasets (MNIST and Olivetti Faces), highlighting the practical implications of our insights for real-world applications.

## 2 Related Work

### 2.1 Training dynamics for GANs

Much of this work is inspired by Wang et al. [6], which was one of the first to undertake this task. However, in this case, they use a single-feature discriminator, and show that the choice of learning rates relative to the strength of noise is what determines the outcome of the training process: convergence, oscillations, or mode-collapse.

There have been further attempts to understand the convergence of GANs through other approaches, not just the dynamics of the gradient-based optimization. Heusel et al. [7] showed that under reasonable assumptions on the training process and hyperparameters, using a specific update rule will guarantee that the GAN converges to a local Nash equilibrium. Mazumdar et al. [8] introduced a new learning algorithm under which the local Nash equilibria are the only attracting fixed points, so that the training algorithm tends to move towards these points. This type of analysis is very similar to our approach, focused on understanding fixed points. Other types of models have also been analyzed in the high-dimensional regime, such as linear VAEs [9], two-layer neural networks solving a regression task [10], and two-layer autoencoders [11]. A review of these methods can be found in [12].

## 2.2 Subspace learning

Subspace learning is a heavily explored field, with many algorithms. However, when the noise of the data has a non-zero variance, most approaches fail, and the general technique used to solve the problem is some type of online PCA-based method. In Wang et al. [13], a similar analysis of dynamics through ODEs is performed for multiple algorithms which learn subspaces with non-zero variance of noise. This analysis allows for steady-state and phase transition analysis of these algorithms. Balzano et al. [14] presents a survey of different online PCA algorithms used for subspace learning, in the case where only some of the data is visible at each timestep, and discuss how it is possible to find a unique subspace of a given rank which matches all the provided data.

# 3 Background and problem formulation

## 3.1 True data model

Our data $\mathbf{y}_k$ is drawn from the following generative model, known as a *spiked covariance model* [15]:

$$\mathbf{y}_k = \mathbf{U}\mathbf{c}_k + \sqrt{\eta_T}\mathbf{a}_k \tag{1}$$

Here, $\mathbf{U} \in \mathbb{R}^{n \times d}$ is the true subspace we wish to learn represented as an orthonormal basis, $\mathbf{c}_k \in \mathbb{R}^d$ is a zero-mean random vector with covariance matrix $\Lambda$, $\mathbf{a}_k \in \mathbb{R}^n$ is a standard Gaussian vector, and $\eta_T$ represents the noise level.

The spiked covariance model is very widely studied, due to the non-triviality of learning $\mathbf{U}$ whenever $\eta_T > 0$. The key property of this model is that the top $d$ eigenvectors of the data covariance $\mathbb{E}[\mathbf{y}_k\mathbf{y}_k^T]$ are given by the columns of $\mathbf{U}$. If there exists a strict eigengap between the top $d$ corresponding eigenvalues and the other eigenvalues, then the reconstruction loss function is proven to have $\mathbf{U}$ as a global minima [16].

## 3.2 Online subspace learning algorithms

Subspace learning is a very important task in machine learning, most commonly performed by algorithms such as PCA or ICA. However, these approaches involve costly operations such as calculating covariance matrices or calculating matrix inverses, infeasible in high dimensions. Therefore, it is very common to use an online version of these algorithms, processing samples one at a time.

Online subspace learning algorithms typically fall into two categories: algebraic methods and geometric methods. Algebraic methods are based on computing the top eigenvectors of some representation of a sample covariance matrix. Assuming a strict eigengap, the top eigenvectors will yield the true subspace. Meanwhile, geometric methods optimize a certain loss function over some geometric space (Euclidean space or a Grassmannian manifold). We review two subspace learning algorithms here, Oja's Method [1] and GROUSE [2]. While GROUSE was introduced for the missing data case, it can be used for full data too. For further details about these algorithms and their categorization, we direct the reader to [14].

However, we suggest a third category of online subspace learning algorithms, which we call the generative methods. Such methods, including single-layer GANs, do not have information about the specific task, and instead aim to learn the data simply by seeing the data and attempting to generate data from the same distribution.

### 3.2.1 Oja's method

Oja's method [1] is a classical algebraic approach to online subspace learning. Given an orthonormalized initial matrix $\mathbf{X}_0$, we perform the following update at every timestep given a data sample $\mathbf{y}_k$:

$$\mathbf{X}_{k+1} = \Pi\left[\mathbf{X}_k + \tau\mathbf{y}_k\mathbf{y}_k^T\mathbf{X}_k\right] \tag{2}$$

Here, $\Pi$ is an orthonormalization operator, and $\tau$ is the learning rate.

### 3.2.2 GROUSE

GROUSE [2] performs gradient descent on the Grassmannian manifold, which guarantees orthonormality of the updates. In the full data case, we again start with an orthonormal initial matrix $\mathbf{X}_0$, and

at each timestep, given a data sample $y_k$, our update is:

$$\mathbf{X}_{k+1} = \mathbf{X}_k + (\cos\theta_k - 1)\frac{\mathbf{p}_k}{||\mathbf{p}_k||}\frac{\mathbf{w}_k^T}{||\mathbf{w}_k||} + \sin\theta_k\frac{\mathbf{r}_k}{||\mathbf{r}_k||}\frac{\mathbf{w}_k^T}{||\mathbf{w}_k||} \tag{3}$$

Here, $\theta_k = \tau||\mathbf{r}_k||||\mathbf{p}_k||$, $\mathbf{w}_k = \arg\min_{\mathbf{w}}||\mathbf{y}_k - \mathbf{X}_k\mathbf{w}||_2^2$, $\mathbf{p}_k = \mathbf{X}_k\mathbf{w}_k$, $\mathbf{r}_k = \mathbf{y}_k - \mathbf{p}_k$, and $\tau$ is our learning rate.

### 3.3 Generative Models

Here, we focus specifically on GANs. A GAN model seeks to learn a representation of the underlying subspace through the use of two components: a generator and a discriminator. The generator learns the subspace by trying to generate new samples from the subspace, while the discriminator acts as a classifier, attempting to distinguish data from the true subspace from data produced by the generator.

Note that measuring performance through cosine similarity can actually be viewed as a way to measure the generalization performance of the generator, as it doesn't depend on any specific instance of generated data and instead provides a concrete measure of how similar the generated data will be.

#### 3.3.1 Generator

We assume that the generator also follows a spiked covariance model:

$$\tilde{\mathbf{y}}_k = \mathbf{V}_k\tilde{\mathbf{c}}_k + \sqrt{\eta_G}\tilde{\mathbf{a}}_k \tag{4}$$

However, we do not assume that $\eta_G = \eta_T$, or that the covariance of $\tilde{\mathbf{c}}_k$, $\tilde{\Lambda}$, is the same as $\Lambda$. The goal of the generator is to learn $\mathbf{V}_k$.

#### 3.3.2 Discriminator

The learning in the GAN model critically depends on the choice of discriminator, which aims to separate the data from the true and generated subspaces.

The most common approach when training GANs is to use a discriminator that is weaker than the generator. If the discriminator is too strong, then it will easily learn to distinguish between true and generated samples, leading to vanishing gradients for the generator and thus preventing learning. However, a weak discriminator results in sequential learning, where the generator is only able to learn a subset of the features at a time. In multi-feature cases, this will lead to very slow learning.

Motivated by this, we seek to analyze a model in which the discriminator has the same strength as the generator. Thus, we let $\mathbf{W} \in \mathbb{R}^{n \times d}$, and define the discriminator as

$$\mathcal{D}(\mathbf{y}; \mathbf{W}) = \hat{\mathcal{D}}(\mathbf{y}^T\mathbf{W}) \tag{5}$$

where $\hat{\mathcal{D}} : \mathbb{R}^n \to \mathbb{R}$ is some function (see the assumptions below). Since this discriminator is able to focus on all the features at once, this means the generator is also able to learn every feature at once. This is in contrast to the single-feature case (where $\mathbf{W} \in \mathbb{R}^n$) analyzed previously. While this is a strong assumption on the discriminator, we show below how this assumption can be relaxed.

#### 3.3.3 Training procedure

GAN training is modeled as a two-player minimax game, where the discriminator attempts to maximize some loss function and the generator attempts to minimize it. This is used as a way to learn a "surrogate" subspace which represents the true subspace. Therefore, the GAN model can be seen as a form of subspace learning, except that the focus is on generating new samples from the subspace.

Specifically, let $\mathcal{L}(\mathbf{y}, \tilde{\mathbf{y}}; \mathbf{W})$ be a loss function depending on the discriminator weights, and true and fake samples. If $\mathcal{G}$ denotes the true distribution and $\tilde{\mathcal{G}}$ denotes the generator distribution, the minimax game can be represented as

$$\min_{\mathbf{V}} \max_{\mathbf{W}} \mathbb{E}_{y\sim\mathcal{G}}\mathbb{E}_{\tilde{y}\sim\tilde{\mathcal{G}}}\mathcal{L}(\mathbf{y}, \tilde{\mathbf{y}}; \mathbf{W}).$$

Following the approach of Wang et al. [6], and in order to compare the sequential and multi-feature cases, we use the following loss function:

$$\mathcal{L}(\mathbf{y}, \tilde{\mathbf{y}}; \mathbf{W}) = F(\hat{D}(\mathbf{y}^T\mathbf{W})) - \hat{F}(\hat{D}(\tilde{\mathbf{y}}^T\mathbf{W})) - \frac{\lambda}{2}tr(H(\mathbf{W}^T\mathbf{W})) + \frac{\lambda}{2}tr(H(\mathbf{V}^T\mathbf{V})) \tag{6}$$

Here, $F, \hat{F}$ are functions affecting the outputs of the discriminator, $H$ is an element-wise function used for regularizing the weights of the generator and discriminator, and $\lambda > 0$ controls the strength of the regularization. As $\lambda \to \infty$, the matrices $\mathbf{V}, \mathbf{W}$ will become orthonormal.

The standard approach to solve this minimax game is using stochastic gradient descent (SGD). At timestep $k$, given a sample $\mathbf{y}_k$ from the true subspace and a sample $\tilde{\mathbf{y}}_k$ from the generator subspace, we perform the following updates:

$$
\begin{aligned}
\mathbf{V}_{k+1} &= \mathbf{V}_k - \frac{\tilde{\tau}}{n} \nabla_{\mathbf{V}_k} \mathcal{L}(\mathbf{y}_k, \tilde{\mathbf{y}}_k; \mathbf{W}_k), \\
\mathbf{W}_{k+1} &= \mathbf{W}_k + \frac{\tau}{n} \nabla_{\mathbf{W}_k} \mathcal{L}(\mathbf{y}_k, \tilde{\mathbf{y}}_k; \mathbf{W}_k).
\end{aligned}
\tag{7}
$$

Here, $\tau$ denotes the learning rate of the discriminator, and $\tilde{\tau}$ denotes the learning rate of the generator. Note that while it is common to use a batch of data at a time when using SGD, we focus on a single element at a time in order to simplify all the analysis.

## 4  Development of ODE

Similar to [6], we make the following definitions:

**Definition 4.1.** $\mathbf{X}_k := [\mathbf{U}, \mathbf{V}_k, \mathbf{W}_k] \in \mathbb{R}^{n \times 3d}$ is called the *microscopic state* of the training process at time $k$.

**Definition 4.2.** The tuple $\{\mathbf{P}_k, \mathbf{Q}_k, \mathbf{R}_k, \mathbf{S}_k, \mathbf{Z}_k\}$ is called the *macroscopic state* of $\mathbf{X}_k$ at time $k$, where $\mathbf{P}_k := \mathbf{U}_k^T \mathbf{V}_k$, $\mathbf{Q}_k := \mathbf{U}^T \mathbf{W}_k$, $\mathbf{R}_k := \mathbf{V}_k^T \mathbf{W}_k$, $\mathbf{S}_k := \mathbf{V}_k^T \mathbf{V}_k$, and $\mathbf{Z}_k := \mathbf{W}_k^T \mathbf{W}_k$. The macroscopic state can be written in matrix notation as $\mathbf{M}_k = \mathbf{X}_k^T \mathbf{X}_k$, in which we get

$$
\mathbf{M}_k = \begin{bmatrix} \mathbf{I} & \mathbf{P}_k & \mathbf{Q}_k \\ \mathbf{P}_k^T & \mathbf{S}_k & \mathbf{R}_k \\ \mathbf{Q}_k^T & \mathbf{R}_k^T & \mathbf{Z}_k \end{bmatrix}.
\tag{8}
$$

### 4.1  Macroscopic dynamics

To analyze the macroscopic dynamics, we reduce to a special case, which leads to a slightly modified set of the assumptions from Wang et al. [6].

(A.1) The sequences $\mathbf{c}_k, \tilde{\mathbf{c}}_k$ are i.i.d. random variables with bounded moments of all orders, and $\{\mathbf{c}_k\}$ is independent of $\{\tilde{\mathbf{c}}_k\}$.

(A.2) The sequences $\{\mathbf{a}_k\}, \{\tilde{\mathbf{a}}_k\}$ are both independent Gaussian vectors with zero mean and covariance matrix $I_n$.

(A.3) $H(\mathbf{A}) = \log \cosh \mathbf{A} - \mathbf{I}$, $\hat{D}(\mathbf{x}) = \|x\|$, and $F(x) = \hat{F}(x) = \frac{x^2}{2}$. We note that the first derivative of $H$ exists, the first four derivatives of $F(\hat{D}(\cdot)), \hat{F}(\hat{D}(\cdot))$ exist, and all the derivatives are uniformly bounded. Thus, our choices satisfy the conditions of assumption (A.3) from Wang et al. [6].

(A.4) Let $[\mathbf{U}, \mathbf{V}_0, \mathbf{W}_0]$ be the initial microscopic state. For $i = 1, \cdots, n$, we have $\mathbb{E}[\sum_{l=1}^d ([\mathbf{U}]_{i,l}^4 + [\mathbf{V}_0]_{i,l}^4 + [\mathbf{W}_0]_{i,l}^4)] \leq C/n^2$, where $C$ is some constant not depending on $n$.

(A.5) The initial macroscopic state $\mathbf{M}_0$ satisfies $\mathbb{E}\|\mathbf{M}_0 - \mathbf{M}_0^*\| \leq C/\sqrt{n}$, where $\mathbf{M}_0^*$ is a deterministic matrix and $C$ is some constant not depending on $n$.

(A.6) The columns of the discriminator matrix $\mathbf{W}$ are orthonormal, so that $\mathbf{W}^T \mathbf{W} = \mathbf{I}_d$.

Assumptions (A1) and (A2) are the usual i.i.d assumptions common in machine learning. (A3) is important for deriving the update equations. (A4) and (A5) are used to guarantee that the macroscopic state can converge. Our assumption (A6) of orthonormal discriminator matrix allows us to simplify the equations since the $\mathbf{Z}$ matrix of the macroscopic state is always just $\mathbf{I}_d$.

Under these assumptions, as well as letting $\lambda \to \infty$, we obtain a modified Theorem 1 from Wang et al. [6], specifically considering the reduced case of equation (13). Note that our choice of $F, \tilde{F}, \hat{D}$ means that our equations become an arbitrary-dimensional version of the original equations.

**Theorem 4.3.** *Fix $T > 0$. Under Assumptions (A.1) - (A.6), it holds that*

$$\max_{0 \leq k \leq nT} \mathbb{E}\|\boldsymbol{M}_k - \boldsymbol{M}(\frac{k}{n})\| \leq \frac{C(T)}{\sqrt{n}}, \tag{9}$$

*where $C(T)$ is some constant depending on $T$ but not $n$, and $\boldsymbol{M}(t) : \mathbb{R}_+ \cup \{0\} \to \mathbb{R}^{3d \times 3d}$ is a deterministic function. Moreover, $\boldsymbol{M}(t)$ is the unique solution of the following ODE:*

$$\frac{d}{dt}\boldsymbol{P}_t = \tilde{\tau}(\boldsymbol{Q}_t\boldsymbol{R}_t^T\tilde{\Lambda} + \boldsymbol{P}_t\boldsymbol{L}_t), \quad \frac{d}{dt}\boldsymbol{Q}_t = \tau(\Lambda\boldsymbol{Q}_t - \boldsymbol{P}_t\tilde{\Lambda}\boldsymbol{R}_t + \boldsymbol{H}_t\boldsymbol{Q}_t)$$

$$\frac{d}{dt}\boldsymbol{R}_t = \tau(\boldsymbol{P}_t^T\Lambda\boldsymbol{Q}_t - \boldsymbol{S}_t\tilde{\Lambda}\boldsymbol{R}_t + \boldsymbol{H}_t\boldsymbol{R}_t) + \tilde{\tau}(\tilde{\Lambda} + \boldsymbol{L}_t)\boldsymbol{R}_t \tag{10}$$

$$\frac{d}{dt}\boldsymbol{S}_t = \tilde{\tau}(\boldsymbol{R}_t\boldsymbol{R}_t^T\tilde{\Lambda} + \tilde{\Lambda}\boldsymbol{R}_t\boldsymbol{R}_t^T + \boldsymbol{S}_t\boldsymbol{L}_t + \boldsymbol{L}_t\boldsymbol{S}_t), \quad \frac{d}{dt}\boldsymbol{Z}_t = \boldsymbol{0}$$

*with the initial condition $\boldsymbol{M}(0) = \boldsymbol{M}_0^*$, where*

$$\boldsymbol{L}_t = -diag(\boldsymbol{R}_t\boldsymbol{R}_t^T\tilde{\Lambda}), \quad \boldsymbol{H}_t = (1 - \frac{\tau\eta_G}{2})\boldsymbol{R}_t^T\tilde{\Lambda}\boldsymbol{R}_t - (1 + \frac{\tau\eta_T}{2})\boldsymbol{Q}_t^T\Lambda\boldsymbol{Q}_t - \tau\frac{\eta_G^2 + \eta_T^2}{2}\boldsymbol{I}. \tag{11}$$

A sketch of the proof of this theorem can be found in Appendix B. The proof closely mirrors the proof of the original theorem in [6].

## 4.2 Microscopic dynamics

The microscopic dynamics are concerned with how the terms $\mathbf{U}, \mathbf{V}, \mathbf{W}$ change over time. Following previous work, we consider the empirical measure

$$\mu_k(\mathbf{U}, \mathbf{V}, \mathbf{W}) = \frac{1}{n}\sum_{i=1}^{n} \delta([\hat{\mathbf{u}}, \hat{\mathbf{v}}, \hat{\mathbf{w}}] - \sqrt{n}[[\mathbf{U}]_{i,:}, [\mathbf{V}_k]_{i,:}, [\mathbf{W}_k]_{i,:}]). \tag{12}$$

where $\delta$ is the delta measure. This is a discrete-time stochastic process, which can be embedded in continuous time as $\mu_t^{(n)} = \mu_k$, with $k = \lfloor nt \rfloor$. Then, as $n \to \infty$, this process converges to a deterministic process $\mu_t$, which is the measure of the solution of the SDE

$$d\hat{\mathbf{u}}_t = \mathbf{0}, \quad d\hat{\mathbf{v}}_t = \tilde{\tau}(\hat{\mathbf{w}}_t\tilde{\Lambda}\mathbf{R}_t + \mathbf{L}_t\hat{\mathbf{v}}_t)dt,$$
$$d\hat{\mathbf{w}}_t = \tau(\hat{\mathbf{u}}^T\Lambda\mathbf{Q}_t + \hat{\mathbf{w}}\mathbf{h}_t)dt + \tau AdB_t, \tag{13}$$

where $A$ is some diffusion term, negligable due to our assumption on the discriminator (A.6).

From this equation and the convergence of the measure, we can obtain the following weak PDE

$$\frac{d}{dt}\langle\mu_t, \varphi(\hat{\mathbf{u}}, \hat{\mathbf{v}}, \hat{\mathbf{w}})\rangle = \tilde{\tau}\left\langle\mu_t, \left(\hat{\mathbf{w}}_t\tilde{\Lambda}\mathbf{R}_t + \mathbf{L}_t\hat{\mathbf{v}}_t\right)\nabla_{\hat{\mathbf{v}}}\varphi\right\rangle + \tau\left\langle\mu_t, \left(\hat{\mathbf{u}}^T\Lambda\mathbf{Q}_t + \hat{\mathbf{w}}\mathbf{h}_t\right)\nabla_{\hat{\mathbf{w}}}\varphi\right\rangle \tag{14}$$

where $\varphi$ is a bounded, smooth test function. The ODE in the main theorem can be derived from this weak PDE.

## 5 Simulations

In order to demonstrate that the ODE properly represents the training dynamics of the GAN model, we first perform simulations and show that the empirical results match the ODE, seen in Figure 1. To understand how the training dynamics change based on the generator learning rate, we fix the discriminator learning rate as $\tau = 0.2$ and fix the generator learning rate $\tilde{\tau} = 0.04$. We show the results on 4 different noise levels. In all cases, we let $\Lambda = \tilde{\Lambda} = diag([\sqrt{3}, \sqrt{5}])$.

We set $\mathbf{P}_0 = \mathbf{Q}_0 = 0.1 * I$, and we ensure that the empirical setup is initialized with exactly matching $P$ and $Q$ values. We note that the ODE will never learn when the initialization is exactly 0, and so we must provide some level of similarity to start training. However, this is not very restrictive, as our experiments show that even random matrices will have approximately $0.001 * I$ for both $P$ and $Q$, which is sufficient to escape the fixed point around 0.

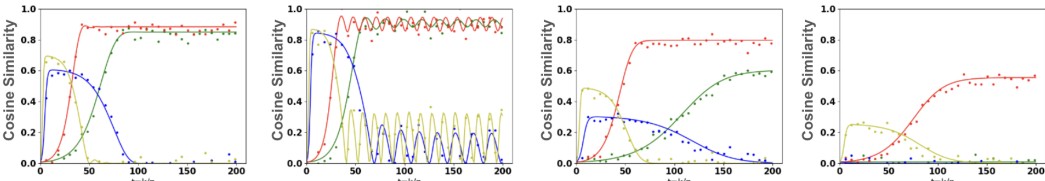

Figure 1: ODE results for learning rate $\tilde{\tau} = 0.04, \tau = 0.2$ and four different noise levels, with $d = 2$. The columns represent $\eta_G = \eta_T = 2, 1, 3, 4$ respectively. At $\eta = 5$ or higher, the generator is unable to learn anything. In all cases, the green and red represent the two diagonals of $\mathbf{P}$, and the blue and yellow represent the two diagonals of $\mathbf{Q}$. We see that the simulations do match the predicted ODE results.

## 5.1 Off-diagonal simulations

A key insight found from the multi-feature discriminator is that the interaction between different features can help learning. When the macroscopic states are initialized to non-diagonal matrices, we see that the dimension with smaller covariance is actually able to attain better results and reach a similar cosine similarity to the dimension with higher covariance. Such an outcome is not possible in the sequential learning regime, due to the lack of interaction between features. In sequential learning, features are learned one at a time, and once a feature has been learned, the training will focus on a different feature instead. This phenomenon can be seen in Figure 2, showing that the off-diagonal initialization allows for not only faster training (which also happens in the diagonal initialization case), but also higher steady-state values compared to the sequential learning case. We are unable to provide a detailed characterization of these fixed points, as a neat closed-form solution cannot be obtained.

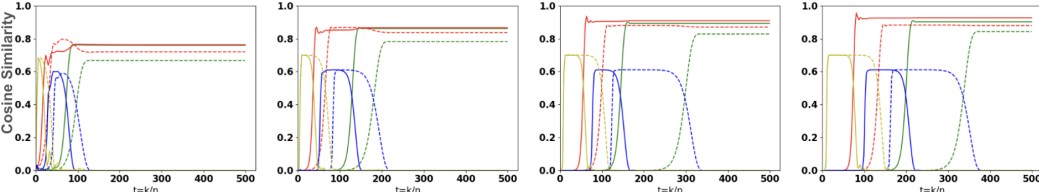

Figure 2: ODE results when initialized with off-diagonal entries. We focus on the case $\eta_G = \eta_T = 2$, as that noise level is seen above to be ideal for learning. Additionally, in all cases, $\tilde{\tau} = 0.04, \tau = 0.2$. The solid lines are with our approach, while the dashed lines are using the discriminator in Wang et al. [6]. From left to right, we use an initialization of $0.1, 0.01, 0.001, 0.0001$ for each component of the macroscopic states. It can be seen that our approach outperforms the single-feature discriminator in every case, with the gap becoming larger as the initialization approaches $0$.

## 6 Unknown number of features

While this type of analysis can provide interesting insights, it has a very restrictive assumption that we know the number of features $d$. This is done so that the macroscopic states are well-defined. However, we now seek to extend this analysis to the case where the true subspace has $d$ features, the generator subspace has $p$ features, and the discriminator learns $q$ features, where we do not assume that $d = p = q$. While this analysis can be performed under any assumptions on the relative size of $d$, $p$, and $q$, we focus on the single case $d \leq q \leq p \ll n$.

To simplify the demonstration of this approach, we make the assumption that $\mathbf{U} = \begin{bmatrix} \mathbf{I}_d \\ \mathbf{0} \end{bmatrix}$, so that $\mathbf{U}$ contains the first $d$ standard basis vectors. We introduce the idea of uplifting (inspired by the work in [17]) the matrices $\mathbf{U}, \mathbf{W}$ to the dimensionality of $\mathbf{V}$.

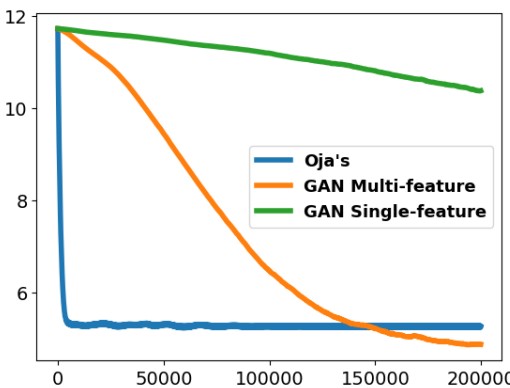

Figure 3: The graph shows the Grassmann distance over time on the Olivetti Faces dataset, for Oja's method (Blue) and the GAN model (Orange), as well as the single-feature GAN model (Green). We use the same hyperparameters as all previous experiments, measured with respect to a full PCA decomposition which acts as a surrogate for the true subspace.

First, since $\mathbf{U}$ is an orthonormal matrix, it lives in the Grassmannian $Gr(d, n)$ of $d$-dimensional subspaces of $\mathbb{R}^n$. Similarly, $\mathbf{W} \in Gr(q, n)$. Our goal is to embed $\mathbf{U}$ and $\mathbf{W}$ into $Gr(p, n)$. Once we do this, we can again calculate the macroscopic states we are interested in. To do this, we use the following map:

$$\mathbf{U} = \begin{bmatrix} \mathbf{I}_d \\ \mathbf{0}_{n-d} \end{bmatrix} \mapsto \begin{bmatrix} \mathbf{I}_d & \mathbf{0}_{n-p \times p-d} \\ \mathbf{0}_{n-d \times d} & \mathbf{I}_p \end{bmatrix}. \tag{15}$$

This produces a new matrix $\bar{\mathbf{U}} \in Gr(p, n)$. We can perform a similar trick with $\mathbf{W}$ to obtain a matrix $\bar{W}$. The important details about this uplifting trick are the following: (1) Due to the construction, we preserve orthonormality of all the matrices, (2) the subspaces of interest are found as the first $d$ columns of the matrix $\bar{\mathbf{U}}$ and the first $q$ columns of the matrix $\bar{\mathbf{W}}$, and (3) the analysis of the diagonal case is unchanged under this uplifting (In the diagonal case, there is no interaction between the different dimensions, so we ignore the other dimensions. In the non-diagonal case, these additional dimensions only provide minor noise, and so don't affect the training at all).

## 7  Real image subspace learning

In order to demonstrate the practicality of this analysis, we test our approach on the MNIST [18] and Olivetti Faces [19] dataset, and compare our approach with the single-feature discriminator from Wang et al. [6]. Here, we include some qualitative results regarding the learned features, and provide a quantitative analysis on the performance differences between the multi-feature and single-feature discriminators. We include the Olivetti Faces results in Figure 4, and the MNIST results can be found in Appendix A.

To perform these visualizations and measure performance, we first perform PCA on the entire dataset and extract the top $K$ (16 or 36) features. We then use this as an approximation of the true subspace $\mathbf{U}$, which allows us to compare the distances. We then track the Grassmann distance between the true and learned subspaces for both the multi-feature and single-feature approaches. The Grassmann distance between two $d$-dimensional subspaces of an $n$-dimensional space is given by

$$d(\mathbf{U}, \mathbf{V}) = \left( \sum_{i=1}^{d} \theta_i^2 \right)^{1/2}, \tag{16}$$

where the $\theta_i$ are the principal angles between the subspaces. Here, a lower distance means a better similarity between the subspaces. If the two matrices are orthonormal, the principal angles are the singular values of the cosine similarity matrix, explicitly connected with the macroscopic states.Figure 3 shows the Grassmann distances for the sequential and multi-feature learning cases on

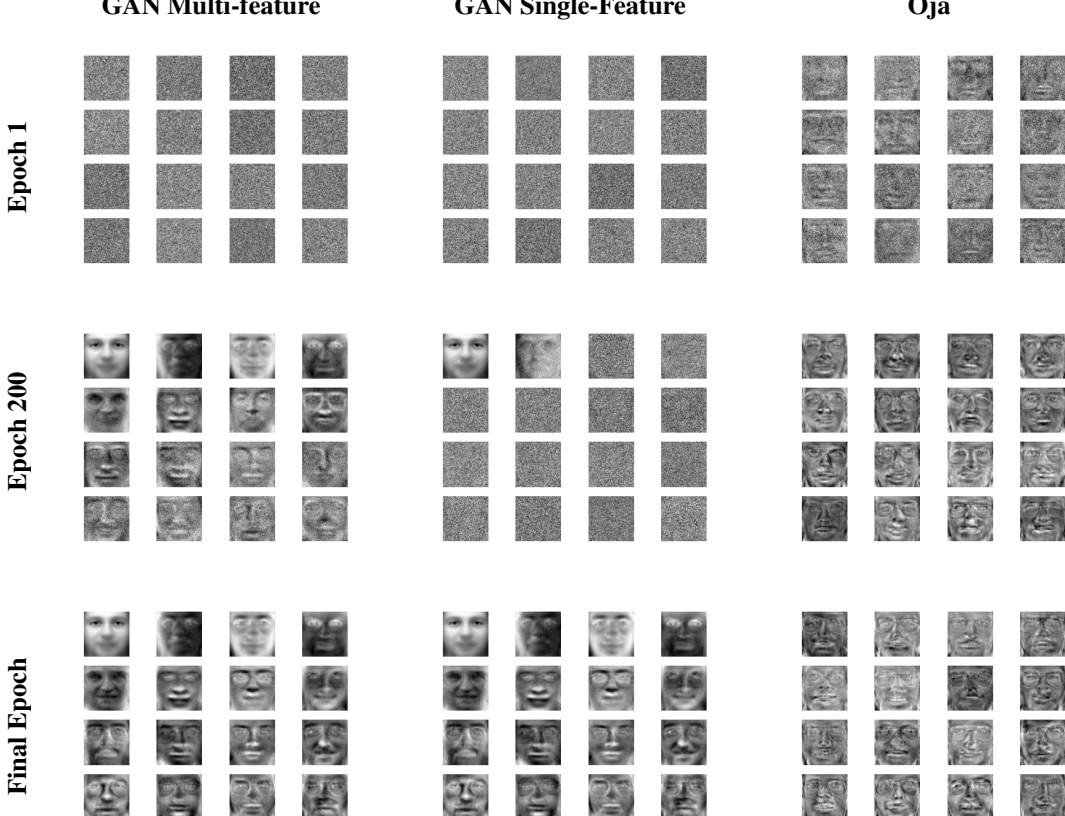

Figure 4: We provide results on the Olivetti Faces dataset, a well-known dataset. We show the top 16 learned features for all approaches at 3 stages of training: after the 1st epoch, the 200th epoch, and the end of training. We train all approaches for 500 epochs, equivalent to approximately 50 timesteps of simulated training. It can be clearly seen that while Oja's method learns quicker than the GAN model, eventually the GAN model outperforms it. Additionally, we see that the features learned by the GAN model are much more diverse and meaningful than those learned by Oja's method (whose learned features are more similar). For the single-feature GAN model, we can see that the learning is significantly slower, and never approaches anywhere close to the other two results.

the Olivetti Faces dataset. This provides empirical justification on a real dataset, showing first that the phenomenon of faster training identified by the ODE in Figure 1 applies to practical settings as well. Furthermore, due to having no restrictions on off-diagonal entries of the macroscopic states, we see that the results in Figure 2 also apply to practical datasets, since our multi-feature discriminator attains better performance even in less time.

## 8   GANs as a subspace learning algorithm

In the linear setting, GANs attempt to perform subspace learning. However, GANs do not fall into either of the categories introduced earlier. The other subspace learning algorithms all seek to minimize the following loss function

$$J(\mathbf{U}) = \mathbb{E}_\mathbf{x}\left[\mathbf{x} - \mathbf{U}\mathbf{U}^T\mathbf{x}\right] \tag{17}$$

known as the reconstruction error. This is because the global optima of this loss function is the true subspace itself, and so, we can view this as a prior included in the subspace learning algorithms. GANs do not have such information, and instead seeks to learn the subspace simply through seeing the datapoints. Therefore, we can consider GANs to be a third type of subspace learning algorithm, which we call the generative algorithms. We seek to understand how well the GAN model is able to

learn a subspace compared to the existing subspace learning algorithms. We compare both analytically using the derived ODEs, as well as empirically on synthetic and the MNIST dataset, in order to see under what circumstances GANs learn a subspace at a comparable rate.

## 8.1 Learned features

Figure 6 in the Appendix compares the features learned by the GAN model to the features learned by Oja's method. Both models are initialized to exactly the same weights, and trained on the same data at the same time, for a single epoch. For the GAN model, we use the same hyperparameters as the previous experiments above. For Oja's method, we used a learning rate of $0.1$, which experimentally we found to produce the best results. We can clearly see that the features learned by the GAN model are more meaningful and more clearly resemble the true data, while most of the features that Oja's method learns aren't very interpretable. This suggests that because the GAN needs to be able to generate the images, this acts as a form of regularization on what types of features are learned.

## 9 Conclusion

Our investigation into single-layer GAN models through the lenses of online subspace learning and scaling limit analysis has provided valuable insights into their data subspace learning dynamics. By extending our analysis to include multi-feature discriminators, we've unearthed novel phenomena pertaining to the interactions among different features, significantly enhancing learning efficiency. This advantage is particularly pronounced in scenarios of near-zero initialization, where the generator achieves higher maximum and steady-state performances compared to the sequential discriminator. Moreover, the interaction between dimensions enables the generator to closely match variances across dimensions, a feat unattainable in the sequential scenario. In the context of subspace learning, we see that in higher noise levels, the GAN is able to more consistently outperform Oja's method on a wide range of generator, discriminator, and Oja learning rates.

Introducing an uplifting method for analysis in arbitrary dimensionalities enables us to better model uncertainties inherent in real-world subspace modeling. Practical validation on the MNIST and Olivetti Faces datasets reaffirms the applicability of our theoretical findings, underscoring the superiority of overparametrization in single-layer GANs over data availability. This prompts intriguing avenues for research in multi-layer GANs, probing whether similar phenomena persist in more complex architectures. Exploring these directions holds promise for further advancements in the field. Finally, we observe that GAN models excel in acquiring a more meaningful feature basis compared to Oja's method when applied to the real-world datasets, which we attribute to their ability to generate new data samples.

## Acknowledgements

We acknowledge that this work was supported in part by TUBITAK 2232 International Fellowship for Outstanding Researchers Award (No. 118C337) and an AI Fellowship provided by Koç University & İş Bank Artificial Intelligence (KUIS AI) Research Center.

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

# A MNIST results

## A.1 Comparisons with Single-Feature Discriminator and Oja's Method

In order to demonstrate that our approach works in more practical settings, we train our model on the MNIST dataset. Then, in order to understand what the model has learned about the dataset, we compute the SVD of the generator weights $\mathbf{V}$, and plot the left singular vectors. Each of these vectors correspond to a single feature learned by the model, and so viewing these will help understand the model performance. Finally, we perform the same tests using the single-feature discriminator, to demonstrate the effects of sequential vs non-sequential learning of features.

Our theory and development in the paper operated under the assumption that we knew everything about the true subspace. While this is not possible for these image datasets (since we cannot determine the true subspace $\mathbf{U}$ or the distribution of $\mathbf{c}$), we can still use the same assumptions and model structure. Therefore, the generator still samples a $\tilde{\mathbf{c}}$ from a standard Gaussian distribution, and the choice of covariance and noise levels are determined through testing.

The dataset is flattened into a $1 \times 784$ vector, so our ambient dimension $n = 784$. For our multi-feature model, we train for a single epoch. For the sequential discriminator, we train for 5 epochs. We focus on the $d = 36$ case, although it can be further scaled up as necessary. Through testing, we fix the covariance matrix $\tilde{\Lambda} = 5 * \mathbf{I}_d$ and $\eta_G = 1$. We use a generator learning rate of $\tilde{\tau} = 0.04$ and a discriminator learning rate of $\tau = 0.2$. While the multi-feature discriminator is able to learn good representations of all 36 basis elements as seen in Figure 5, the sequential discriminator is unable to learn even half of them in the 5 epochs. As can be seen, the last 18 basis elements are just noise.

This scaling becomes very problematic as the number of features increases. Even with just 36 features, a small amount given modern datasets, such a model requires significantly more training and is still unable to perform as well as the multi-feature model.

Finally, we provide a comparison of the GAN learned features with the Oja's learned features in Figure 6. It can be seen that most of the GAN features are more visually representative of the dataset compared to Oja's method.

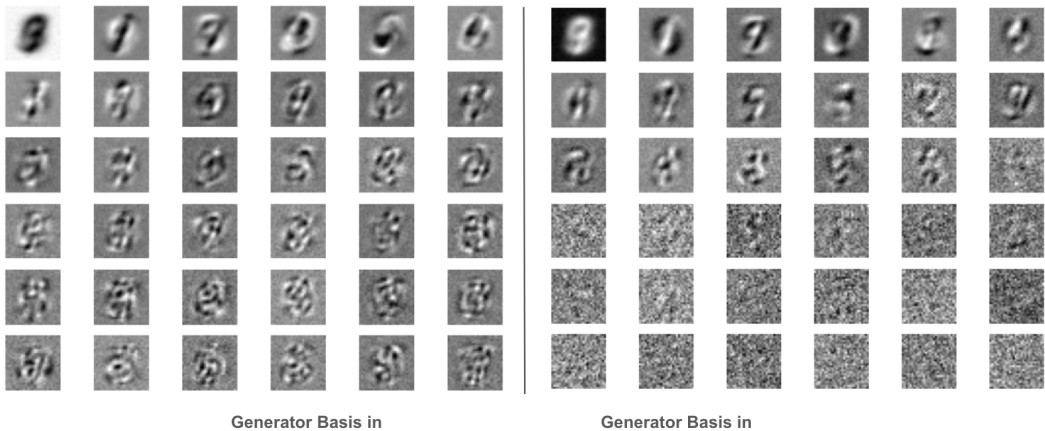

**Generator Basis in Multi-Feature Case**   **Generator Basis in Single-Feature Case**

Figure 5: Comparison between the generator basis vectors learned by the multi-feature and single-feature discriminators on 36 features. The multi-feature model is trained for 1 epoch, while the single-feature model is trained for 5 epochs.

## A.2 Grassmann distances

Figure 7 contains a comparison of the Grassmann distances for the multi-feature and single-feature cases. Even after 5 epochs, the sequential discriminator still has a much higher Grassmann distance than the multi-feature model, even though it has seen 5 times as much data. Specifically, after one epoch of training, the multifeature discriminator has a distance of 2.46, while the single-feature

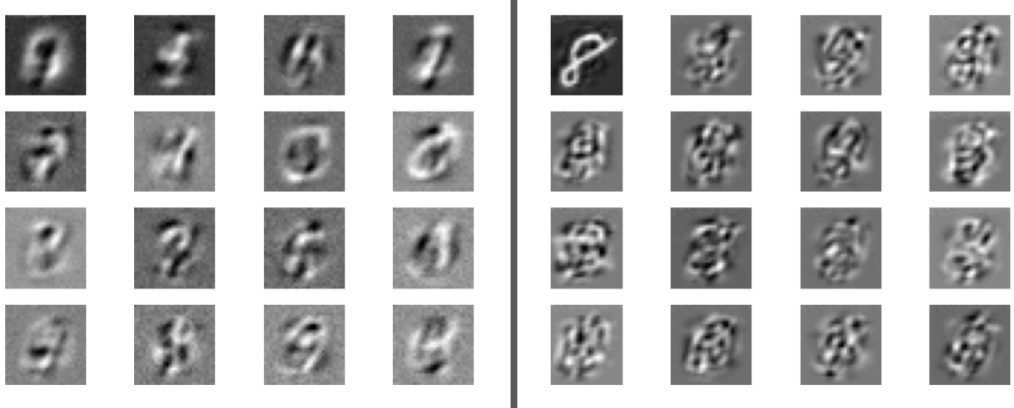

**GAN Learned Features**                **Oja Learned Features**

Figure 6: After training both GAN and Oja's method on the MNIST dataset with 16 features, we use SVD to extract their learned features, and compare them here. We can see that despite Oja's method learning quicker than the GAN model (seen in previous analysis), the GAN features learned are a better representation of the data, while most of Oja's features do not resemble the data. Note that Oja learning an $8$ as the first feature is due to the order of training samples seen.

discriminator finishes with a distance of $3.17$, showing a significant gap. We tested with up to $20$ epochs, but saw no improvements for the sequential discriminator past $5$ epochs.

We also see an example of the training outcomes predicted by the ODE in Figure 7. Specifically, our choice of learning rates $\tilde{\tau} = 0.04, \tau = 0.2$ and noise level $\eta = 1$ is seen in Row 1, Column 2 of Figure 1, and we see the expected result of the generator oscillating around its steady state.

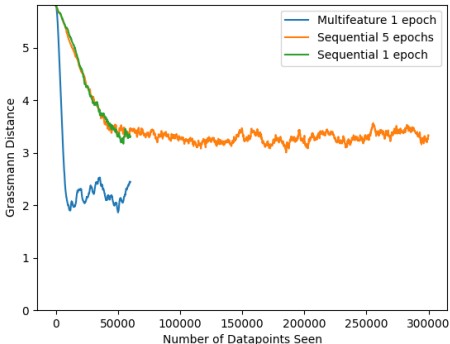

Figure 7: In the first figure, using PCA on the MNIST dataset, we obtain the top 16 features of the data, and use these features as an approximation for the true subspace $\mathbf{U}$. We then track the Grassmann distance between the learned subspace and this approximation, for both our model trained for 1 epoch, and the sequential discriminator trained for 1 and 5 epochs. It is clearly seen that our discriminator learns much faster than the sequential discriminator, while at the same time obtaining a much lower distance, even when the sequential discriminator has sees $5$ times as much data.

# B  Proof of main theorem

This proof mirrors the proof of Theorem 1 in [6]. However, for completeness, we re-state the key results and sketch the proof here.

The proof relies on the following result, found in [13]:

**Lemma B.1.** *Consider a sequence of stochastic processes $\{\boldsymbol{x}_k^{(n)}, k = 0, 1, 2, \cdots, \lfloor nT \rfloor\}_{n=1,2,\cdots}$ with some constant $T > 0$. If $\boldsymbol{x}_k^{(n)}$ can be decomposed into three parts*

$$\boldsymbol{x}_{k+1}^{(n)} - \boldsymbol{x}_k^{(n)} = \frac{1}{n}\phi(\boldsymbol{x}_k^{(n)}) + \rho_k^{(n)} + \delta_k^{(n)}, \tag{18}$$

*such that*

*(C.1) The process $\sum_{k'=0}^{k} \rho_{k'}^{(n)}$ is a martingale, and $\mathbb{E}||\rho_k^{(n)}||^2 \le C(T)/n^{1+\epsilon_1}$ for some $\epsilon_1 > 0$;*

*(C.2) $\mathbb{E}||\delta_k^{(n)}|| \le C(T)/n^{1+\epsilon_2}$ for some $\epsilon_2 > 0$;*

*(C.3) $\phi(\boldsymbol{x})$ is a Lipschitz function, i.e., $||\phi(\boldsymbol{x}) - \phi(\tilde{\boldsymbol{x}}) \le C||\boldsymbol{x} - \tilde{\boldsymbol{x}}||$;*

*(C.4) $\mathbb{E}||\boldsymbol{x}_k^{(n)}||^2 \le C$ for all $k \le \lfloor nT \rfloor$;*

*(C.5) $\mathbb{E}||\boldsymbol{x}_0^{(n)} - \boldsymbol{x}_0^*|| \le C/n^{\epsilon_3}$ for some $\epsilon_3 > 0$ and deterministic vector $\boldsymbol{x}_0^*$,*

*then we have*

$$||\boldsymbol{x}_k^{(n)} - \boldsymbol{x}(\frac{k}{n})|| \le C(T)n^{-\min(\frac{1}{2}\epsilon_1, \epsilon_2, \epsilon_3)}, \tag{19}$$

*where $\boldsymbol{x}(t)$ is the solution of the ODE*

$$\frac{d}{dt}\boldsymbol{x}(t) = \phi(\boldsymbol{x}(t)) \quad \text{with } \boldsymbol{x}(0) = \boldsymbol{x}_0^*. \tag{20}$$

The relevant stochastic process is the macroscopic states introduced in Section 4. The macroscopic states are decomposed as

$$\mathbf{M}_{k+1} - \mathbf{M}_k = \frac{1}{n}\phi(\mathbf{M}_k) + (\mathbf{M}_{k+1} - \mathbb{E}_k \mathbf{M}_{k+1}) + \left[\mathbb{E}_k \mathbf{M}_{k+1} - \mathbf{M}_k - \frac{1}{n}\phi(\mathbf{M}_k)\right]. \tag{21}$$

Note that our macroscopic state can just be written as an $n^2$ dimensional vector, and it is equivalent to using the Frobenius norm in the conditions above. We seek to show that this decomposition satisfies the conditions (C.1) - (C.5).

Immediately, Condition (C.5) is satisfied by the assumption (A.5) for the theorem. Additionally, (C.3) is satisfied by assumption (A.3) and Lemma 4 in the supplementary material of [6].

Next, we slightly modify Lemmas 2 and 7 in the supplementary material of [6] for our case.

**Lemma B.2** (Lemma 7 of [6]). *Under the assumptions (A.1) - (A.6), given $T > 0$, we have*

$$||\mathbb{E}_k \boldsymbol{v}_{k+1,i} - \boldsymbol{v}_{k,i}|| \le Cn^{-1}(||\boldsymbol{v}_{k,i}|| + ||\boldsymbol{w}_{k,i}||),$$
$$||\mathbb{E}_k \boldsymbol{w}_{k+1,i} - \boldsymbol{w}_{k,i}|| \le Cn^{-1}(||\boldsymbol{u}_i|| + ||\boldsymbol{v}_{k,i}|| + ||\boldsymbol{w}_{k,i}||). \tag{22}$$

The proof of this follows exactly from Lemma 7 of [6].

**Lemma B.3** (Lemma 2 of [6]). *Under the assumptions (A.1) - (A.6), given $T > 0$, we have*

$$\mathbb{E}\left(\sum_{l=1}^{d}[\boldsymbol{V}_k]_{i,l}^4 + [\boldsymbol{W}_k]_{i,l}^4\right) \le C(T)n^{-2}. \tag{23}$$

*Proof.* We show that this holds for a fixed $i$.
First, we know that

$$\mathbb{E}_k||\mathbf{w}_{k+1,i} - \mathbf{w}_{k,i}||^{\gamma} \le \frac{C}{n^{\gamma}}(1 + ||\mathbf{u}_i||^{\gamma} + ||\boldsymbol{v}_{k,i}||^{\gamma} + ||\boldsymbol{w}_{k,i}||^{\gamma}). \tag{24}$$

whenever $\gamma = 2, 3, 4$, due to boundedness of $h, f, \tilde{f}$. Additionally, we can write

$$\begin{aligned}
\mathbb{E}[\mathbf{w}_{k+1}]_{i,l}^4 - \mathbb{E}[\mathbf{w}_k]_{i,l}^4 = {} & 4\mathbb{E}\left[[\mathbf{w}_k]_{i,l}^3\mathbb{E}([\mathbf{w}_{k+1}]_{i,l} - [\mathbf{w}_k]_{i,l})\right], \\
& + 6\mathbb{E}\left[[\mathbf{w}_k]_{i,l}^2\mathbb{E}([\mathbf{w}_{k+1}]_{i,l} - [\mathbf{w}_k]_{i,l})^2\right], \\
& + 4\mathbb{E}\left[[\mathbf{w}_k]_{i,l}\mathbb{E}([\mathbf{w}_{k+1}]_{i,l} - [\mathbf{w}_k]_{i,l})^3\right], \\
& + \mathbb{E}\mathbb{E}_k([\mathbf{w}_{k+1}]_{i,l} - [\mathbf{w}_k]_{i,l})^4.
\end{aligned} \tag{25}$$

Combining both of these, we get that

$$\mathbb{E}[\mathbf{w}_{k+1}]_{i,l}^4 - \mathbb{E}[\mathbf{w}_k]_{i,l}^4 \leq \frac{C}{n}\left(n^{-2} + \mathbb{E}||\mathbf{u}_i||^4 + \mathbb{E}||\mathbf{v}_{k,i}||^4 + \mathbb{E}||\mathbf{w}_{k,i}||^4\right),$$

$$\leq \frac{C}{n}\mathbb{E}(n^{-2} + \sum_{l=1}^d [\mathbf{V}_k]_{i,l}^4 + [\mathbf{W}_k]_{i,l}^4). \tag{26}$$

which follows from assumption (A.4). Similarly, we get

$$\sum_{l=1}^d \mathbb{E}([\mathbf{V}_{k+1}]_{i,l}^4 - [\mathbf{V}_k]_{i,l}^4) \leq \frac{C}{n}\mathbb{E}(n^{-2} + \sum_{l=1}^d [\mathbf{V}_k]_{i,l}^4 + [\mathbf{W}_k]_{i,l}^4). \tag{27}$$

Combining these for both terms and iteratively applying it, we get

$$\mathbb{E}(\sum_{l=1}^d [\mathbf{V}_k]_{i,l}^4 + [\mathbf{W}_k]_{i,l}^4) \leq (n^{-2} + \sum_{l=1}^d [\mathbf{V}_0]_{i,l}^4 + [\mathbf{W}_0]_{i,l}^4)e^{\frac{k}{n}C}. \tag{28}$$

Then, due to assumption (A.4), we get the required result. □

Once we have Lemmas B.2 and B.3, we can show that condition (C.4) is satisfied.

**Lemma B.4.** *Condition (C.4) is satisfied for our macroscopic state stochastic process.*

*Proof.* We show that the expected norm squared of each macroscopic state is less than some $C(T)$. The cases of $\mathbf{P}_k$ and $\mathbf{S}_k$ are proven in Lemma 3 of [6], and require no changes. Additionally, by our assumption (A.6) that the matrix $\mathbf{W}_k$ is orthonormalized, we know that $\mathbf{Z}_k = \mathbb{I}_d$, and so the requirement is trivially satisfied for $\mathbf{Z}_k$. Thus, it remains to show this for $\mathbf{Q}_k$ and $\mathbf{R}_k$. We show this for $\mathbf{Q}_k$, and $\mathbf{R}_k$ follows similarly.

$$\mathbb{E}[\mathbf{Q}_k]_{l,l'}^2 = \mathbb{E}(\sum_{i=1}^n [\mathbf{U}]_{i,l}[\mathbf{W}_k]_{i,l'})^2,$$

$$\leq \mathbb{E}(\sum_{i=1}^n [\mathbf{U}]_{i,l}^2)\mathbb{E}(\sum_{i=1}^n [\mathbf{W}]_{i,l'}^2),$$

$$\leq \sqrt{\mathbb{E}(\sum_{i=1}^n [\mathbf{U}]_{i,l}^2)^2 \mathbb{E}(\sum_{i=1}^n [\mathbf{W}]_{i,l'}^2)^2}, \tag{29}$$

$$\leq C(T),$$

as required. □

**Lemma B.5.** *Condition (C.2) above is satisfied, meaning that for all $k = 0, 1, \cdots, \lfloor nT \rfloor$, and for a given $T > 0$, we have*

$$\mathbb{E}||\mathbb{E}_k\boldsymbol{M}_{k+1} - \boldsymbol{M}_k - \frac{1}{n}\phi(\boldsymbol{M}_k)|| \leq C(T)n^{-3/2}. \tag{30}$$

*Proof.* To prove this, we can split it into five parts, one for each of the macroscopic states. For the macroscopic state $\mathbf{Z}_k$, this just requires showing that

$$\mathbb{E}||\mathbb{E}_k\mathbf{Z}_{k+1} - \mathbf{Z}_k|| \leq C(T)n^{-3/2}. \tag{31}$$

But the left side is just zero, since $\mathbf{Z}_k = \mathbb{I}_d$ for all $k$. Thus, this is trivially satisfied.
For the macroscopic state $\mathbf{P}_k$, we want to show that

$$\mathbb{E}||\mathbb{E}_k\mathbf{P}_{k+1} - \mathbf{P}_k - \frac{\tilde{\tau}}{n}(\mathbf{Q}_k\mathbf{R}_T\tilde{\Lambda} + \mathbf{P}_k\mathbf{L}_k)|| \leq C(T)n^{-3/2}. \tag{32}$$

However, from the gradient of our update equation for $V$, averaging over $\tilde{c}_k, \tilde{a}_k$ we see that

$$\mathbf{E}_k\mathbf{V}_{k+1} - \mathbf{V}_k = \frac{\tilde{\tau}}{n}\left[\mathbf{W}_k\mathbf{R}_k^T\tilde{\Lambda} + \mathbf{V}_k\mathbf{L}_k\right]. \tag{33}$$

Multiplying both sides by $\mathbf{U}^T$ on the left, we get

$$\mathbb{E}_k \mathbf{P}_{k+1} - \mathbf{P}_k = \frac{\tilde{\tau}}{n} \left[ \mathbf{Q}_k \mathbf{R}_k^T \tilde{\Lambda} + \mathbf{P}_k \mathbf{L}_k \right]. \tag{34}$$

But then, the left side of the equation we wanted to show is just zero, and so the inequality is satisfied. Applying a similar process to the update equation for $W_k$, we want to show that

$$\mathbb{E}||\mathbb{E}_k \mathbf{Q}_{k+1} - \mathbf{Q}_k - \frac{\tau}{n}(\Lambda \mathbf{Q}_k - \mathbf{P}_k \tilde{\Lambda} \mathbf{R}_k + \mathbf{H}_k \mathbf{Q}_k)|| \leq C(T) n^{-3/2}, \tag{35}$$

and by averaging over $\tilde{\mathbf{c}}_k, \tilde{\mathbf{a}}_k, \mathbf{c}_k, \mathbf{a}_k$ and multiplying by $\mathbf{U}^T$ on the left, we get

$$\mathbb{E}_k \mathbf{Q}_{k+1} - \mathbf{Q}_k = \frac{\tau}{n} \left[ \Lambda \mathbf{Q}_k - \mathbf{P}_k \tilde{\Lambda} \mathbf{R}_k + \mathbf{H}_k \mathbf{Q}_k \right]. \tag{36}$$

Again, this results in the left side of the expression we want to show just being zero.
Finally, we show the result for $\mathbf{S}_k$. The case for $\mathbf{R}_k$ follows similarly to the previous results.
Using the property that

$$\mathbf{S}_{k+1} - \mathbf{S}_k = \mathbf{V}_k^T(\mathbf{V}_{k+1} - \mathbf{V}_k) + (\mathbf{V}_{k+1} - \mathbf{V}_k)^T \mathbf{V}_k + (\mathbf{V}_{k+1} - \mathbf{V}_k)^T(\mathbf{V}_{k+1} - \mathbf{V}_k), \tag{37}$$

and averaging over $\tilde{\mathbf{c}}_k, \tilde{\mathbf{a}}_k$, we get

$$\mathbb{E}_k \mathbf{S}_{k+1} - \mathbf{S}_k = \frac{\tilde{\tau}}{n} \left[ \mathbf{R}_k \mathbf{R}_k^T \tilde{\Lambda} + \mathbf{S}_k \mathbf{L}_k + \tilde{\Lambda} \mathbf{R}_k \mathbf{R}_k^T + \mathbf{L}_k \mathbf{S}_k \right] + \frac{\tilde{\tau}^2}{n^2} \left[ \mathbf{W}_k \mathbf{R}_k^T \tilde{\Lambda} + \mathbf{V}_k \mathbf{L}_k \right]^T \left[ \mathbf{W}_k \mathbf{R}_k^T \tilde{\Lambda} + \mathbf{V}_k \mathbf{L}_k \right]. \tag{38}$$

The second term in the sum above has expected norm

$$\mathbb{E}||\frac{\tilde{\tau}^2}{n^2} \left[ \mathbf{W}_k \mathbf{R}_k^T \tilde{\Lambda} + \mathbf{V}_k \mathbf{L}_k \right]^T \left[ \mathbf{W}_k \mathbf{R}_k^T \tilde{\Lambda} + \mathbf{V}_k \mathbf{L}_k \right]|| \leq \mathbb{E}||\mathbf{W}_k \mathbf{R}_k^T \tilde{\Lambda} + \mathbf{V}_k \mathbf{L}_k||^2,$$

$$\leq 2||\mathbf{Z}_k||||\mathbf{R}_k^T \tilde{\Lambda}||^2 + 2||\mathbf{S}_k||||\mathbf{L}_k||^2, \tag{39}$$

$$\leq C\mathbb{E}[||\mathbf{Z}_k|| + ||\mathbf{S}_k||],$$

$$\leq C(T).$$

This concludes the proof. □

For condition (C.1), the requirement of being a martingale is automatically satisfied by construction. To show that the remainder of condition (C.1) is satisfied, it suffices to prove that

$$\mathbb{E}||\mathbf{M}_{k+1} - \mathbf{M}_k||^2 \leq C(T) n^{-2}. \tag{40}$$

**Lemma B.6.**

$$\mathbb{E}||\mathbf{M}_{k+1} - \mathbf{M}_k||^2 \leq C(T) n^{-2}. \tag{41}$$

*Proof.* We can break this up into each of the 5 macroscopic states separately. As before, doing this for $\mathbf{Z}_k$ is trivial. We show this for $\mathbf{P}_k$ and $\mathbf{Q}_k$, and the rest follow similarly. For $\mathbf{P}_k$, we get

$$\mathbb{E}||\mathbf{P}_{k+1} - \mathbf{P}_k||^2 \leq C n^{-2} \mathbb{E}\left[||\mathbf{Q}_k||\mathbb{E}_k||\tilde{\mathbf{c}}_{2k}^2 + ||\mathbf{P}_k||^2\right],$$

$$\leq C n^{-2} \mathbb{E}\left[1 + ||\mathbf{Q}_k||^2 + ||\mathbf{P}_k||^2\right], \tag{42}$$

$$\leq C(T) n^{-2}.$$

This finishes the proof for $\mathbf{P}_k$.
For $\mathbf{Q}_k$, we have

$$\mathbb{E}||\mathbf{Q}_{k+1} - \mathbf{Q}_k||^2,$$

$$\leq \frac{\tau^2}{n^2} \mathbb{E}\left[||\mathbf{c}_k||^2 f_k^2 + ||\mathbf{U}^T \mathbf{a}_k||^2 f_k^2 + ||\mathbf{P}_k||^2 ||\tilde{\mathbf{c}}_{2k}||^2 \tilde{f}_{2k}^2 + ||\mathbf{U}^T \tilde{\mathbf{a}}_{2k}||^2 \tilde{f}_{2k}^2 + ||\mathbf{Q}_k||^2 ||\mathbf{H}_k||^2\right],$$

$$\leq C n^{-2} \left[1 + \sqrt{\mathbb{E}||\mathbf{U}^T \mathbf{a}_k||^4} \sqrt{\mathbb{E} f_k^4} + \sqrt{\mathbb{E}||\mathbf{U}^T \tilde{\mathbf{a}}_k||^4} \sqrt{\mathbb{E} \tilde{f}_k^4} + \mathbb{E}||\mathbf{Z}_k||^2 + \mathbb{E}||\mathbf{S}_k||^2\right], \tag{43}$$

$$\leq C n^{-2} \left[1 + \mathbb{E}||\mathbf{Z}_k||^2 + \mathbb{E}||\mathbf{S}_k||^2\right],$$

$$\leq C(T) n^{-2}.$$

where in the last line, we used the previously calculated values for $\mathbb{E}||\mathbf{Z}_k||^2$ and $\mathbb{E}||\mathbf{S}_k||^2$. The values $f_k$ and $\tilde{f}_{2k}$ are the values of $f = F'$ and $\tilde{f} = \tilde{F}'$ evaluated on the corresponding inputs.
The conditions for the rest of the macroscopic states can be shown in the same way. □

Given the previous lemmas, the proof of the theorem then follows immediately from Lemma B.1.

