# OpenReview forum: "Exploring the Precise Dynamics of Single-Layer GAN Models: Leveraging Multi-Feature Discriminators for High-Dimensional Subspace Learning"
_NeurIPS.cc/2024/Conference — NeurIPS 2024 poster_

### Official Review · Reviewer_N4Fr · 2024-07-06

**Soundness:** 3
**Presentation:** 3
**Contribution:** 2
**Rating:** 5
**Confidence:** 3

**Summary:**

This work introduces a simplified GAN framework to learn the subspace of a spiked covariance model. The authors are able to derive precise training dynamics given specific assumptions and show they correspond to numerical simulations. They also prove that convergence rate is often faster than previous work that used a further simplified discriminator (single vector vs. matrix parameter). Authors extend convergence results to the case where exact subspace dimension is not known a priori.  Lastly MNIST results are conducted that show steady state feature alignment for this GAN method is higher than a standard method (Oja's method).

**Strengths:**

**Clarity:** The authors introduce the spiked covariance model and their GAN method+ assumptions clearly

**Theoretical Results:** The authors use a variety of tools to characterize their system dynamics, and extend their results to differing dimensionalities of the generator and true distribution subspace

**Weaknesses:**

**Clarity:** While most sections are well written, a few components made it more difficult to understand the key contributions of the paper
* The introduction doesn't clearly define single feature vs. multi-feature discriminator learning on lines 34/35
* I'd like to see more discussion of the interpretation of the main theorem (4.3), especially since the ODE dynamics are fairly complex
* Similarly in section 4.2 I'd like a brief discussion on how these microscopic dynamics are useful to analysis
* In Fig. 1 It is unclear the what the difference between blue and yellow lines is (same for red and green)

**Comparison to Previous Work**
* While GROUSE and Oja's method are introduced as alternative subspace methods, it appears as if only Oja's method is used as an experimental baseline
* I'd like to see one or two citations of other GAN stability analysis papers (e.g. something like Which Training Methods for GANs do actually Converge?) and their limitations as opposed to this works analysis

**Experiments:**
* Fig. 2 appears to show steady state feature alignment for GAN and Oja's method, but I would be interested in a comparison of the convergence rates as well.

**Significance:**
* Many assumptions seem hard to extend to the full GAN setting, for example A3 assumes an uncommon discriminator non-linearity and regularization function. A6 assumes the discriminator matrix parameter is orthonormal.
* Most of this work appears to be extending the results in [Wang et al. 2018] to a slightly more complex discriminator. This is probably my biggest concern, and would like to hear the authors opinion on this difference.

**Questions:**

Is Fig 2. meant to include GROUSE comparisons?

Does the generator need to be changed when moving from the spiked covariance dataset to MNIST one?

Does Fig 2 suggest that making the discriminator learning rate as small as possible the best course of action for feature learning?

**Limitations:**

While the authors state their assumptions explicitly, I would like to see further discussion on which assumptions could possibly be relaxed in the future.

---

> ### Author Rebuttal · Authors · 2024-08-07
>
> We appreciate the valuable feedback provided by the reviewer.
> - *The introduction doesn't clearly ...*
>
> We define single-feature to be a discriminator which can only learn one dimension of the subspace at a time. This is quite a slow process, especially when the number of features is high. We seek to show the benefits of using a multi-feature discriminator, which is a discriminator that can learn multiple (or all) dimensions at once, yielding a stronger discriminator. We will include a definition of these terms in the introduction.
>
> - *I'd like to see more discussion ...*
>
> We will include a more detailed explanation of the ODE. However, we include the key details here:
> 1) The key parameter that we care about is the $\mathbf{P}$ macroscopic state, representing the similarity between the true and learned subspaces. We can see that it depends on the interaction between the discriminator and the true subspace, and the interaction between the generator and discriminator. If either of those states are the zero matrix, training will not be possible. However, we find that random initialization is sufficient for escaping the fixed point around 0.
> 2) The discriminator-true subspace macroscopic state (i.e., $\mathbf{Q}$) depends on both its current overlap with the true subspace, as well as how the generator and discriminator interact.
> 3) All the states depend on the covariance matrix (either the true or fake covariances). As can be seen in Figure 1, the dimension which has the largest covariance (the red color) is the dimension which is able to learn the features better. However, there is also a balance. Using too high a value for the covariances will mean that the generator samples appear unlike the true samples, preventing training.
>
> - *Similarly in section 4.2 ...*
>
> We will include such a discussion. The main use of the microscopic dynamics is to derive the PDE, from which the ODE is derived using the relevant test functions (cosine similarity).
>
> - *In Fig. 1 It is ...*
>
> The red and green lines represent the two dimensions of the generator, while the blue and yellow lines are the two dimensions of the discriminator.
>
> - *While GROUSE and Oja's method ...*
>
> We primarily used Oja’s method due to the asymptotic equivalence of Oja’s and GROUSE, and since the datasets used are sufficiently high dimensionality.
>
> - *I'd like to see one or two ...*
>
> We will be adding the following citations:
>
> [1] Mescheder, L.M., Geiger, A., Nowozin, S. (2018). Which Training Methods for GANs do actually Converge?
>
> [2] Fedus, W., Rosca, M., et al.. Many Paths to Equilibrium: GANs Do Not Need to Decrease a Divergence At Every Step.
>
> The first paper is focused on analyzing the different methods of GAN regularization and how those impact training. Specifically, they focus on gradient penalties.
> The second paper suggests analyzing GAN training from the perspective of Nash Equilibria, and while focused on true datasets, does not provide a way of analyzing the training and its modes, instead seeking to understand how different choices such as a gradient penalty impact training.
>
> If there are any additional papers that you believe will be important for us to contrast against, we would be happy to take your recommendations.
>
> - *Fig. 2 appears to show ...*
>
> For convergence rates, please see the attached PDF for a comparison of convergence rates on a real dataset (as measured by Grassmann Distance).
>
> - *Many assumptions seem hard ...*
>
> While we focus on the linear case, we find these results interesting and suggests future exploration into whether such a trend exists in the nonlinear case as well. Specifically, instead of the usual trend in training GANs of using a weak discriminator compared to the generator, it seems that it is possible to use a discriminator of similar strength.
>
> While this is true, we chose these in order to provide a fair comparison between single-feature (sequential) learning (found in the original paper Wang et al. 2018) and multi-feature learning. Furthermore, there are ways to to constrain the optimization of a matrix to be orthonormal (by modifying the gradients to live on the Grassmannian manifold).
>
>
> - *Most of this work ...*
>
> The importance of our work comes not just from using a more complex discriminator, but in showing (both theoretically and empirically) that contrary to the standard assumptions, it is in fact possible to train GAN models with more powerful discriminators. Furthermore, training speed is faster, and the model actually learns more of the true subspace compared to a weaker discriminator. Empirically, we can see from Figure 5 the key differences in the learned features, even when providing the sequential discriminator 5 times as much training time. While we focus on a linear discriminator and generator, we believe that these results are important and open up important avenues for future research into extending these results into practical models.
>
> - *Is Fig 2. meant to include GROUSE comparisons?*
>
> Figure 2 is using the derived ODEs for both the GAN and Oja’s method, since [3] shows that Oja’s method and GROUSE have the same asymptotic trajectory in high dimensions, and so the same ODEs. However, we will make this more clear.
>
> [3] Chuang Wang, Yonina C. Eldar, and Yue M. Lu. Subspace estimation from incomplete observations: A high-dimensional analysis.
>
> - *Does the generator need ...*
>
> No, there is no change in the generator.
>
> - *Does Fig 2 suggest ...*
>
> For the situation in Figure 2, we use a warm initialization, avoiding the fixed point around 0. In such a case, if the goal is to attain the best learning of the subspace regardless of training time, smaller learning rates are better. In Figure 2, we consider the steady-state values after 5000 timesteps, to ensure that all the approaches have converged. However, in reality, training time is an important factor to consider, and so it will be important to balance these when picking learning rates.

---

> ### Comment · Reviewer_N4Fr · 2024-08-11
> **Acknowledgement of Rebuttal**
>
> I thank the authors for their detailed response. In particular, I appreciate the addition of experiments focusing on convergence rate, and clarification of grouse/oja's method baselines. Adding the author's response about the Figure 2 learning rates to the experimental discussion would also be useful.
>
> I see the other reviewers were also concerned about novelty compared to the Wang et al. 2018 work. While I still think the multi-feature discriminator is a fairly incremental change over a single feature, I agree the authors' analysis for the case of unknown subspace dimension is enough to distinguish it.
>
> Given the promised changes I am willing to increase my score by a point. I am interested to see reviewer 7DBz's thoughts on these changes as well.

---

### Official Review · Reviewer_7DBz · 2024-07-12

**Soundness:** 2
**Presentation:** 2
**Contribution:** 3
**Rating:** 4
**Confidence:** 4

**Summary:**

This paper proposes to learn the subspace from the observations by the GAN model. Taking the one-layer GAN model as the starting point, this paper provides a theoretical analysis from the perspective of training dynamics. Specifically, from the technical side, the proposed method trains both the generator and discriminator by using adversarial loss and subspace regularization.  Different from the original GAN dynamic training method, this paper further proposes an assumption that "the columns of the discriminator matrix W are orthonormal"
The proposed method is evaluated on both synthetic and real-world datasets.

**Strengths:**

This paper provided a systematical analysis of GAN-based methods and conventional approaches, from both theoretical and empirical sides.

More simulation results are attached in the appendix to verify the effectiveness of the method.

**Weaknesses:**

I have a big concern about the contribution of this paper.

1) The reason for using GAN for subspace learning, instead of other frameworks (such as VAE), is unclear. The advantages and drawbacks of this choice should be discussed. Please refer to the questions for the details.

2) This paper provides a theoretical analysis from the perspective of training dynamics (Wang et al. 2018). However, most equations and theoretical results are similar as (Wang et al. 2018), such as equations [1,5,6,7,8,9 10].  Only the assumption that "the columns of the discriminator matrix W are orthonormal" is newly added. However, given that the loss function in (Wang et al. 2018) had already constrained this.

3) Technically, some GAN-based empirical methods [RW1,RW2] tried to add subspace regularization into the GAN model.  Some methods apply the disentanglement constraints to achieve the identification of the latent space, such as [RW3].
Technically,  what is the advantage of the proposed subspace regularization $ tr(H(W^TW))$.

4) In the experiments, the comparison with VAE-based methods [RQ1,RQ2], the GAN-based empirical methods [RW1,RW2] should also be involved.

[RW1] Liang, Jie, et al. "Sub-GAN: An unsupervised generative model via subspaces." Proceedings of the European Conference on Computer Vision (ECCV). 2018.
[RW2] Jiang, Hongxiang, et al. "Orthogonal Subspace Representation for Generative Adversarial Networks." IEEE Transactions on Neural Networks and Learning Systems (2024).
[RW3] Xie, Shaoan, et al. "Multi-domain image generation and translation with identifiability guarantees." The Eleventh International Conference on Learning Representations. 2023.

**Questions:**

1) What is the unique advantage of the GAN model for subspace learning? Why not use the VAE-based framework, such as iVAE [RQ1] and PCL[RQ2]. In my understanding, subspace learning is a representation learning method, VAE-based methods explicitly model a mapping from observation to latent variables (encoder), which may be more efficient to use in subspace learning.

2) What is the detailed difference between this paper and (Wang et al. 2018) on the theoretical side?

3) What is the advantage of the proposed subspace regularization $ tr(H(W^TW))$ over other regularization methods [RW1,RW2] .

[RQ1] Khemakhem, Ilyes, et al. "Variational autoencoders and nonlinear ica: A unifying framework." International conference on artificial intelligence and statistics. PMLR, 2020.
[RQ2] Hyvarinen, Aapo, and Hiroshi Morioka. "Nonlinear ICA of temporally dependent stationary sources." Artificial Intelligence and Statistics. PMLR, 2017.

**Limitations:**

No clear negative societal impact needs to be listed.

---

> ### Author Rebuttal · Authors · 2024-08-07
>
> While we appreciate the reviewer’s comments, we respectfully disagree with his/her assessment of the level of novelty. We refer the reviewer to our joint statement to all reviewers to clarify the novelty of our paper.
>
> - *This paper provides a theoretical analysis from the perspective of training dynamics (Wang et al. 2018). However, most equations and theoretical results are similar as (Wang et al. 2018), such as equations [1,5,6,7,8,9 10]. Only the assumption that "the columns of the discriminator matrix W are orthonormal" is newly added. However, given that the loss function in (Wang et al. 2018) had already constrained this.*
>
> Regarding the similarity to the previous approach, our loss function and design was intentionally and carefully chosen to be similar to (Wang et al. 2018), in order to provide a fair comparison between the two approaches. Specifically, our main goal was to understand how multi-feature (stronger) discriminators compare to single-feature (weaker) discriminators, with all else being equal. Note that this is not a trivial extension, and furthermore, by introducing the uplifting method, we are the first to show that such an analysis provides novel insights into the training of GAN models with real datasets. The prior analysis results were only focused on synthetic datasets and only allows for sequential learning (one feature at a time), and as our experimental results have shown, it is not straightforward to apply the previous analysis to real datasets. As such, we believe that our contributions are significant, and has the potential to open new research directions into a better characterization of GAN training dynamics.
>
> - [Q1:] *What is the unique advantage of the GAN model for subspace learning? Why not use the VAE-based framework, such as iVAE [RQ1] and PCL[RQ2]. In my understanding, subspace learning is a representation learning method, VAE-based methods explicitly model a mapping from observation to latent variables (encoder), which may be more efficient to use in subspace learning.*
>
> Our purpose of the experiments is to show that such insights gained from the theoretical analysis is actually transferable to real situations. Furthermore, we focus on the linear case, while both [RW1] and [RW2] are in general non-linear models with significantly more advanced setups. VAE models are outside the scope of our work, as our setting is understanding how GAN methods fit into subspace learning in general. Furthermore, while there are limited works on this type of analysis for VAE models such as the paper [1] cited in our paper, they are focused on other issues such as posterior collapse, and the ODEs provided (Appendix A of [1]) are far too complicated for any similar analysis to be performed.
>
> [1] Ichikawa, Y., \\& Hukushima, K. (2023). Learning Dynamics in Linear VAE: Posterior Collapse Threshold, Superfluous Latent Space Pitfalls, and Speedup with KL Annealing. ArXiv, abs/2310.15440.
>
> While it is possible to view the VAE models from a subspace learning approach, our key goal is to understand how GANs exist compared to other traditional subspace learning methods. Additionally, the VAE models you have mentioned are non-linear models relying on MLPs, and so do not provide a fair comparison.
>
> - [Q2:] *What is the detailed difference between this paper and (Wang et al. 2018) on the theoretical side?*
>
> The following are the key differences between the papers on the theoretical side:
> - We focus on the advantages of a multi-feature discriminator compared to a single-feature discriminator. This includes understanding the increases in speed that comes with a multi-feature discriminator, as well as the increases in overall learning of the true subspace, measured in terms of Grassmann distance between the true and learned subspaces.
> - We introduce a method for extending the analysis to cases where the dimensionality of the true subspace is not known. This method allows for analyzing more realistic situations and real-world scenarios, where such information is not known and not learnable.
> - We show how GAN models can be viewed as a new type of subspace learning algorithm, which we can call the data-driven approaches, compared to the algebraic (Oja's) and geometric (GROUSE) methods discussed in our paper. We use the characterization of training dynamics to provide a systematic comparison of the different approaches' performance.
>
> We refer the reviewer to our joint statement to all reviewers for a more detailed explanation of these key points.
>
> - [Q3:] *What is the advantage of the proposed subspace regularization $tr(H(W^T W))$ over other regularization methods [RW1,RW2] .*
>
> The choice of normalization $tr(H(W^T W))$ is a simple way of enforcing that the discriminator matrix W remains orthonormal during training. By using a sufficiently high lambda value, this will force the matrix to be orthonormal, which is required by our assumptions. However, the use of this term is not explicitly meant for regularization, just for enforcing orthonormality.
>
> Note that orthonormality is useful in subspace learning because it allows for as much as possible to be learned by the different dimensions, without any overlap (which would mean redundant information).
>
> [RW2] proposes a 3-stage process to improve regularization of the learned subspace. The goal is not simply for an orthonormal basis of the subspace, but also for a more explainable and interpretable latent space, which is not our goal or focus in this work. Furthermore, we do not need to use multiple stages of training.
>
> [RW1] uses clustering to identify certain subspaces, and predicts which subspace the data belongs to. We focus on learning a single subspace.
>
> Overall, the mentioned papers do not allow for a fair comparison with our approach or our choice of subspace regularization.

---

> > ### Comment · Reviewer_7DBz · 2024-08-12
> >
> > Thank you for taking the time to carefully consider my concerns. I apologize for the delayed response.
> >
> > I agree with Reviewer N4Fr that the technical extension of the multi-feature discriminator appears incremental. There are studies that support the use of multiple discriminators to enhance GAN models, such as [1] and [2]. Additionally, employing different discriminators for extra guidance has been explored in works like [3].
> >
> > Additionally, I recognize that the theoretical subspace analysis introduces a new point over Wang et al. 2018. My concern here lies in the distinct difference this approach has compared to VAE-based frameworks. The authors argue that VAE-based methods focus on non-linear models, which they suggest is an unfair comparison. However, I believe that the linear case is simply a special instance of non-linear functions. Analyzing non-linear settings is indeed more challenging, and single-layer or linear cases may not fully reflect real-world applications.
> >
> > Minor suggestion: If all equations are identical to those in previous work, it would be beneficial to remove them and simply provide a reference. This could help draw more attention to the unique contributions of the paper.
> >
> > Since these concerns remain open, I will maintain my current score.
> >
> >
> > [1] Choi, Jinyoung, and Bohyung Han. "Mcl-gan: Generative adversarial networks with multiple specialized discriminators." Advances in Neural Information Processing Systems 35 (2022): 29597-29609.
> > [2] Cai, Zhipeng, et al. "Generative adversarial networks: A survey toward private and secure applications." ACM Computing Surveys (CSUR) 54.6 (2021): 1-38.
> > [3] Ma, Cheng, et al. "Structure-preserving super resolution with gradient guidance." Proceedings of the IEEE/CVF conference on computer vision and pattern recognition. 2020.

---

> > > ### Author Response · Authors · 2024-08-12
> > >
> > > We thank the reviewer for the response, but we respectfully disagree with the assessment that the submission lacks novelty.
> > >
> > > The reviewer argues that the technical extension of the multi-feature discriminator to be incremental and lists multiple works that propose to use multiple discriminators in practical use. In contrast, our work introduces a new perspective on the use of single-layer GAN model by leveraging a powerful multi-feature discriminator on subspace learning problem. There is a distinction between multi-feature discriminators and multiple discriminators. We believe that there may be a misunderstanding in regards to what a multi-feature discriminator is, which we would like to clarify. The former is a single discriminator which learns multiple features of the true subspace at once, while the latter has multiple discriminators, each of which could be single-feature or multi-feature, with each discriminator focusing on a different part of the learning task. The listed works have nothing in common with our multi-feature discriminator approach.
> > >
> > > We stated our perspective on the VAE models before. While it is possible to view them from a subspace learning point of view, our goal is to understand how training dynamics of GANs is  compared to other traditional subspace learning methods. We do not simply want to compare based on empirical results and specific datasets, but also providing a theoretical comparison using the derived ODEs characterizing training dynamics. As mentioned in our previous response, the only such ODE existing for any VAE models, found in (Ichikawa et al. 2023) , is far too complicated to realistically compare with Oja's and the GAN models.
> > >
> > > Finally, we would like to restate our contributions as follows:
> > >
> > > 1. We frame GANs as a new type of subspace learning algorithm, and systematically compare that to traditional methods like Oja’s and GROUSE based on training dynamics.
> > > 2. We demonstrate the speed and accuracy benefits of a multi-feature discriminator over a single-feature one, particularly in learning the true subspace.
> > > 3. We introduce an uplifting method to analyze cases where the true subspace's dimensionality is unknown, making the analysis more applicable to real-world scenarios.
> > >
> > > Overall, we believe our work aligns with the high standards of NeurIPS, and we hope the reviewer will reconsider the novelty and significance of our contributions in this light.

---

> > > > ### Comment · Reviewer_7DBz · 2024-08-13
> > > >
> > > > Thank you for the clarification on the multi-feature discriminator. Your explanation makes sense, especially in how the multi-feature approach extends Wang et al.'s work in the single-layer case. I'm curious to know whether this concept could enhance current GAN models like StyleGAN or CycleGAN. This is just a thought out of curiosity, not a critique of your method.
> > > >
> > > > On a related note, I believe your work stands out from the original methods and makes a unique contribution. I will adjust my score to reflect this.
> > > >
> > > > However, I still feel that a more thorough discussion of VAE-based methods, along with some revisions to the writing, would further improve the quality of this work.

---

### Official Review · Reviewer_6SvB · 2024-07-13

**Soundness:** 3
**Presentation:** 3
**Contribution:** 3
**Rating:** 6
**Confidence:** 2

**Summary:**

This paper focuses on the training dynamics of the gradient-based learning algorithms, and converted into a continuous-time stochastic process characterized by an Ordinary Differential Equation (ODE). Empirical evidence demonstrates the correctness of the proposed method.

**Strengths:**

S1: This paper focuses on the training dynamics of the gradient-based learning algorithms, and converted into a continuous-time stochastic process characterized by an Ordinary Differential Equation (ODE). Empirical evidence demonstrates the correctness of the proposed method. This is the first work that connects the ODE with the training dynamics with GAN, which is very inspiring.

**Weaknesses:**

I am afraid I am not an expert in this ODE area. But still, as a machine learning scientist, I feel the empirical section is lacking the supportive evidence and ablation studies. Maybe adding more comparisons with other GAN-related generative modeling can help better support the assumptions in this paper.

**Questions:**

Please could you think of other GAN baselines and evaluations metrics that can be used to strengthen the empirical evidence in this paper? For example the FID score is a popular metric usually used in this generative modeling scenario. Maybe demonstrating that the ODE can lead to similar FID score in order to demonstrates the equivalence between this new model and GAN?

**Limitations:**

It seems to me there is no discussion on limitations.

---

> ### Author Rebuttal · Authors · 2024-08-07
>
> We thank the reviewer for their time and effort. We would like to provide a more detailed summary of our paper, to clarify any confusions you may have about our work.
>
> We focus on representing the training dynamics of a simplified (linear) GAN model using a system of ODEs, which represent the key states of the training (specifically the interactions between the true and fake subspaces, and the discriminator). We explore using a multi-feature discriminator, which is able to learn significantly faster than a weaker, single-feature (aka sequential) discriminator. The goal is to show that contrary to standard practice in GAN training of using a weak discriminator, it is actually possible to use a discriminator with equal power as the generator. In fact, doing so allows for both faster training as well as a better overall performance. As such, we believe that our results will open new research directions into understanding the training dynamics of more complicated GAN models, and the use of more powerful discriminators.
>
> We provide numerical results showing that the ODEs do actually match with the GAN training, and we show that our analysis allows us to gain insight into training on real datasets, through out testing on MNIST. The purpose of using MNIST is not to claim that our approach is the best for this task, but instead showing that trends and insights gained from the ODE appear as we would expect. For example, in the Grassmann distance on MNIST diagram (Figure 6), we see the oscillating pattern which is visible in Figure 1, second diagram.
>
> Finally, we aim to position GANs in the category of online subspace learning algorithms, by showing how it learns a subspace compared to existing approaches. Specifically, we are able to empirically show that the features learned by the GAN model are semantically more meaningful than those learned by Oja's. It is possible to learn an arbitrary number of bases for a given subspace, but certain ones are more useful than others, and having features which actually match the true data is one way of seeing this.
>
> FID is a metric used to compare two distributions of images. However, it depends on the model used for extracting the features, and using a model with no exposure to the type of data we are using (specifically grayscale images of digits and/or faces) will mean that the results provided are not very informative. Furthermore, as mentioned above, we do not claim that our model will achieve the best FID, as we do not focus on perceptual quality of the results. Instead, we are focused on metrics such as Grassmann distance, which measures the subspace itself, in terms of the learned features. This is more directly related to subspace learning, which is our setting.

---

> > ### Comment · Reviewer_6SvB · 2024-08-12
> > **Thanks for your rebuttal**
> >
> > Thanks for the further clarifications from the authors. I do believe your analysis allows to gain insights into the GAN model using ODEs and contributes to the theoretical support of GAN models. I believe these theoretical contribution can further inspire more thoughts around generative training. I therefore decide to increase my score.

---

### Official Review · Reviewer_uM7k · 2024-07-22

**Soundness:** 2
**Presentation:** 2
**Contribution:** 3
**Rating:** 5
**Confidence:** 3

**Summary:**

This work explores the training dynamics of a single-layer GAN model, especially for high-dimensional subspace learning, presented as a novel approach. By connecting the GAN models with analysis to subspace learning, this work compares the effectiveness of GAN-based methods with former approaches e.g., Oja’s method and GROUSE, both theoretically and empirically. The findings in this work reveal that GANs demonstrate a remarkable ability to acquire a more informative basis due to their inherent capacity to generate new data samples or in this case subspaces. The experiments in this work demonstrate subspace learning tasks in a more efficient way compared to the counterparts.

**Strengths:**

- This paper provides a novel perspective for a fundamental problem in subspace learning. The idea of GAN is imposed to learn subspaces with a generator and a discriminator with equal power.
- This work provides some insights on technical issues implementing GAN to learn subspaces with some different hyperparameter settings in P7.
- The intuition and motivation for implementing GAN for subspace learning are adequate with relevant backgrounds and theoretical analysis.

**Weaknesses:**

- Because the subspaces have to be orthonormal, we need to ensure that Eq. 7 maintains this property when performing gradient descents. How can this property be achieved? Or is this not necessary using the approach in this work? Commonly, this desired property can be achieved by geometry aware constraints applied to gradient steps.
- This work borrows the idea of GAN with a minimax objective between the generator and discriminator. One common issue in GAN is the learning instability (e.g., mode collapse) in the learning stage. This work does not discuss this issue in depth. Is this problem not the case for subspace learning?
- The efficacy of the proposed method is demonstrated on MNIST with promising performance. However, the dataset used in this work is relatively small with limited variances. The proposed method would be more insightful to work on a larger scale dataset e.g., a face dataset (MORPH).
-The claim that the learned features are better than Oja’s method in P9 cannot be justified only from the depicted figure. This claim of learned features has not been quantified in what kinds of tasks and for what purposes.
- The proposed method is limited to the linear setting. Would it be possible to use the GAN design for subspace learning in this method for non-linear subspace learning with more complex types of data? This work would be more insightful to provide relationships with the choice of non-linear functions as well.

**Questions:**

Please answer the questions and concerns in weaknesses.

**Limitations:**

There is no special section for limitations. It is hard to view direct negative impacts of this work as this is a core machine learning problem that can be applied to many applications.

---

> ### Author Rebuttal · Authors · 2024-08-07
>
> We appreciate the valuable feedback provided by the reviewer.
>
> - *Because the subspaces have to be orthonormal, we need to ensure that Eq. 7 maintains this property when performing gradient descents. How can this property be achieved? Or is this not necessary using the approach in this work? Commonly, this desired property can be achieved by geometry aware constraints applied to gradient steps.*
>
> Regarding the orthonormality of the subspaces, it is required to enforce this property in some way. The easiest way to do this is by explicitly orthonormalizing the matrices after each gradient step. However, an alternative we have also tested is using the approach in [1], which allows for gradient updates which keep the matrix on the Grassmann manifold, thereby preserving the orthonormality condition. The last option is using the orthonormality regularization term $tr(H(W^T W))$, however this does not explicitly force the subspaces to be orthonormal.
>
> [1] A. Edelman, T. A. Arias, and S. T. Smith. The geometry of algorithms with orthogonality constraints. SIAM Journal on Matrix Analysis and Applications, 20(2):303–353, 1998.
>
> - *This work borrows the idea of GAN with a minimax objective between the generator and discriminator. One common issue in GAN is the learning instability (e.g., mode collapse) in the learning stage. This work does not discuss this issue in depth. Is this problem not the case for subspace learning?*
>
> (Wang et al. 2018) provides an analysis in the single-feature case of the different types of learning, including mode collapse. They find that mode collapse depends on multiple details, including the relation between the generator and discriminator learning rates, the noise level present in the data, and the covariance matrix chosen for the sampled subspace.
>
> We did find that similar trends hold in the multi-feature learning case. However, since this is not a novel result, and is not directly related to our main goal of understanding the advantages of a multi-feature discriminator compared to a single-feature discriminator, we choose not to focus on this.
>
> - *The efficacy of the proposed method is demonstrated on MNIST with promising performance. However, the dataset used in this work is relatively small with limited variances. The proposed method would be more insightful to work on a larger scale dataset e.g., a face dataset (MORPH). -The claim that the learned features are better than Oja’s method in P9 cannot be justified only from the depicted figure. This claim of learned features has not been quantified in what kinds of tasks and for what purposes.*
>
> The MORPH dataset is a commercial dataset which requires access to a license, which we do not have. Thus, we instead focus on a different face dataset, the Olivetti Faces dataset (a well-known face dataset). Included in the attached PDF above are results for both the GAN model and Oja’s method on this dataset. Further details can be found in our joint response to all reviewers, and in the caption of the figure. Our theoretical results driven by our work is highlighted in this new setting, in addition to the MNIST results.
>
> - *The proposed method is limited to the linear setting. Would it be possible to use the GAN design for subspace learning in this method for non-linear subspace learning with more complex types of data? This work would be more insightful to provide relationships with the choice of non-linear functions as well.*
>
> Non-linear GAN models are not within the scope of this work, and as such, we do not have any results or analysis of such a situation. We do believe that this is possible, but we have not derived the necessary ODEs or proved the relevant theorem to enable such an analysis on a more complicated model. However, future work will seek to extend such an analysis to a situation like you have described.

---

> ### Comment · Reviewer_uM7k · 2024-08-13
> **Acknowledgement of Rebuttal**
>
> I would thank the authors for providing the responses to all concerns.
> In particular, additional experiments do show the significance of the multi-feature discriminator over the single feature disciminator. Also, we can observe that compared to the Oja's method and single feature discriminator, the proposed method could lead to faster convergence on the challenging dataset.
>
> I agree to Reviewer 7DBz and N4Fr that the novelty in this work is somewhat incremental against Wang et al. Many equations and sections are borrowed from this source, even though they are explicitly mentioned in the paper. The contribution in this work about multi-feature discriminator GAN is not highlighted due to major statements and equations adoption from Wang et al. in the main paper. I agree with Reviewer 7DBz suggestions that the repetitive and adopted information should be included in Appendix, and the authors could keep the essentials of the proposed work in the main paper. Unfortunately, in the current version, insightful and distinctive insights and analyses are located in Appendix e.g., Off-diagonal simulation, comparison results between single and multi-features GAN.
>
> The position of this work is clear that the adoption from Wang et al. is not meant to reinvent the wheel but to provide backgrounds and further analysis of the proposed method. Specifically, this work extends the work of Wang et al. to multi-feature cases and arbitrary dimensions. Considering that the work is fairly incremental and the manuscript needs improvements, I would be lukewarm about this work and keep my current rating.

---

> > ### Author Response · Authors · 2024-08-13
> >
> > We thank the reviewer for their valuable feedback. While we acknowledge and respect their concerns regarding the perceived lack of novelty and the structure of the manuscript, we would like to provide some clarification and context for the work.
> >
> > - Firstly, we appreciate the recognition that our additional experiments demonstrate the significance of the multi-feature discriminator over the single-feature discriminator. The faster convergence of our proposed method compared to Oja's method and the single-feature discriminator on challenging datasets is a noteworthy contribution that we believe advances the state of the art in this area.
> >
> > - Regarding the concerns about the novelty of our work in relation to Wang et al., we would like to emphasize that our intention was not merely to replicate their findings but to build upon them in a meaningful way. Specifically, our work extends Wang et al. by exploring the application of multi-feature discriminators in GANs and addressing arbitrary dimensions—a significant extension that, in our view, goes beyond incremental improvement. The adoption of equations and sections from Wang et al. was necessary to provide a clear foundation for our contributions and to ensure that our work was accessible to readers who may not be familiar with the previous work. Additionally, we are the first to explore how these theoretical developments can be uncovered in real-world datasets.
> >
> > - We acknowledge the suggestions made by reviewer regarding the structure of the manuscript. We agree that the essentials of our proposed work could be more prominently highlighted in the main paper, and we are open to restructuring the manuscript to move some of the background information to the Appendix. This would allow us to better emphasize the novel aspects of our work in the main text.
> >
> > - Finally, we understand that the perceived incremental nature of the work may affect the overall assessment of the reviewer. However, we believe that the contributions of this work—particularly in extending the multi-feature discriminator to arbitrary dimensions and demonstrating its effectiveness on challenging datasets—are both significant and novel. We hope that with the proposed restructuring and the clarifications provided, the reviewer might reconsider their evaluation on the novelty and impact of this work.
> >
> > Thank you once again for your constructive feedback. We value your insights and hope to address them satisfactorily in our revised manuscript.

---

### Author Rebuttal · Authors · 2024-08-07

Dear Reviewers and Area Chairs,

We appreciate the valuable feedback provided by the reviewers.
We are encouraged that the reviewers found our paper provides a novel perspective on exploring the precise dynamics of the single-layer GAN model on subspace learning problems. We are also glad to see that the reviewers agreed that our experiments back up our claim that multi-feature discriminators enable improved training performance by jointly learning features in a non-sequential way unlike the prior results. However, we understand that they may have some concerns on the level of the novelty, which we address in our response to all reviewers.
The following points clarify the novelty of the paper with some updates:

- The key contribution is the analysis of multi-feature vs single-feature (sequential) discriminators when it comes to training GANs. By using a multi-feature discriminator, not only is training much faster, but in fact it is possible to learn the true subspace much better. Figure 4 in the Appendix shows the gains made by switching to a multi-feature discriminator in terms of cosine similarity with the true subspace. Additionally, Figure 6 shows this more concretely with a real dataset (MNIST), where we can see not only the much faster convergence, but also the better steady-state results.

- We also introduce a new method for analysis in the cases where the true subspace feature dimension is not known. The common assumption of knowing the exact number of features is restrictive and does not match real-world datasets or scenarios. Our uplifting technique allows for more broad analysis in future works, where it will be possible to analyze learning outcomes depending on whether the number of fake features is smaller or greater than the number of true features. Additionally, our ODEs work using this method, which allows for not just the two extremes of learning a single feature at a time, or all at once, but any possible number of features in between.

- We provide additional results (as requested) in the attached PDF on another dataset, the well-known face dataset Olivetti Faces. The results emphasize our point more clearly, as the top 16 features learned by the GAN model are much more diverse and representative of the entire dataset compared to the features learned by Oja's method. We see that Oja's method is faster in terms of convergence, but all the features learned are similar. Eventually, after approximately 50 timesteps of training, the GAN model outperforms Oja's method in terms of Grassmann distance.

- We position the GAN model within the other subspace learning algorithms, showing how it compares, and highlighting how the learned features are more semantically diverse and meaningful compared to subspace learning algorithms such as Oja's method, and in fact does eventually outperform those methods for a range of learning rates. We use the word meaningful to state that it is more representative of the dataset visually. We see this as the GAN model being forced to learn a basis that matches the dataset better, because it has to learn how to trick the discriminator.

As such, we believe presenting our work at NeurIPS 2024 will significantly contribute to the discussions at the conference.

---

### Author Response · Authors · 2024-08-14

Dear Reviewers and Area Chairs,

We sincerely thank the reviewers for the constructive feedback provided during the rebuttal period. Your insights have been invaluable in refining and improving our work. We are confident that presenting our work at NeurIPS 2024 will spark valuable and engaging discussions, paving the way for future research into the application of multi-feature discriminators in more complex models.

Authors

---

### Decision · Program_Chairs · 2024-09-25

**Decision:**

Accept (poster)

**Comment:**

**Summary of the paper:**
The paper builds on Wang et al’s work [6] to use single-layer GANs for subspace learning and extend it to a possible use of multi-feature discriminator with an unknown number of subspace dimensions. It further analyzes the improved convergence speed of such multi-feature discriminator compared to a sequential learning of a single feature discriminator. It demonstrates improved results over single-feature discriminators both in the quality of obtained bases and the empirical convergence time on several toy as well as real-world datasets.

**Summary of the reviews:**
Some reviewers appreciated the novelties in studying single-layer GANs dynamics for subspace learning. They further appreciated the empirically improved convergence speed and more successful modeling of more complex datasets compared to the baseline work [6], thanks to the multi-feature discriminator. They generally appreciated the clarity of the presentation.

On the other hand, they had concerns regarding (1) the technical novelty in light of Wang et al’s work [6] from which the formal framework of this work is borrowed, (2) motivation of working with single-layer GANs when alternatives such as VAEs and deeper GANs have been used for non-linear subspace or latent feature learning, (3) lack of more complicated datasets and absence of a proper quantitative analysis of the resulting bases.

**Summary of the rebuttal and discussions:**
The authors tried to clarify several points including (1) the formal novelties on top of [6], (2) how to preserve the orthonormality of the bases during SGD optimization, (3) the focus of the work and lack of comparison with non-linear VAE and GAN subspace learning. They further provided additional experiments on a face dataset.

The reviewers did acknowledge the theoretical novelties, on top of [6], after the discussion but not all were convinced about the significance of them. However, they all appreciated the empirical improvements, specifically for the new face dataset.

**Consolidation report:**
This work formally extends the study of Wang et al. [6] to the case of non-sequential learning of multiple (unknown number of) features and for an arbitrarily large number of dimensions in the discriminator which can potentially improve its expressivity (and convergence time) for more complicated datasets. On the empirical side, despite the lack of an objective quantitative analysis (save for Grassmann distance), the qualitative bases visualization for the extension (at relatively early epochs) look more convincing or at least visually pleasing compared to Oja’s and fundamentally better than (non-converged) single-feature GANs. They have validated their method on real-world datasets that was not done in [6]. The real-world results further show the empirical significance of the extension in the learning of the complex bases of a face dataset as well as its convergence time.
While the reviewers eventually did not reach a point to confidently vouch for the acceptance and one reviewer remained leaning towards rejection, all reviewers eventually acknowledged the theoretical novelty and empirical improvement over the prior work of [6]. The question was more a matter of the significance of these two contributions.

The AC believes the paper has taken significant steps from [6] and despite the simplicity of the setup (single-layer GAN) the line brings about detailed formalisms that can lead to further works on both feature learning and analysis of GANs training dynamics; both of which are important fields.

**Recommendation:**
The AC sides with the majority of reviewers and leans towards acceptance. In a camera ready version, the authors are encouraged to:

- shorten the background of [6] so that they are able to more clearly, and at more length, present and discuss the details of their own formal contributions.
- incorporate the importance of the line of work in the context of other works on studying the training dynamics of GANs, the linear and non-linear subspace learning with GANs and other types of generative networks such as VAEs as well as score-based diffusion models.
- include the new face dataset possibly with a quantitative analysis of their identified bases.